

# Inverse modelling for surface methane flux estimation with 4DVar: impact of a computationally efficient representation of a non-diagonal B-matrix in INVICAT v4

Ross N Bannister[1,3] and Chris Wilson[2,3]

[1]Dept. of Meteorology, Univ. of Reading, Whiteknights Road, Earley Gate, Reading, RG6 6ET, UK
[2]School of Earth and Environment, Univ. of Leeds, LS2 9JT, UK
[3]National Centre for Earth Observation

**Correspondence:** Ross N Bannister (r.n.bannister@reading.ac.uk)

**Abstract.**

   Prior information is essential to most inverse problems and the surface flux estimation problem is no exception. The uncertainties of the prior fields, and their inter-correlations, should ideally be reflected in the a-priori error covariance matrix, often called $\mathbf{B}$. The $\mathbf{B}$-matrix, is however, difficult to quantify partly because it is typically a large matrix and partly because its

numerical values are unknown.

   We present a highly efficient method of representing the $\mathbf{B}$-matrix to represent prior errors in the initial concentration and in the time sequence of surface fluxes for the 4DVar-based inverse modelling system (INVICAT) used to estimate the surface fluxes of methane. Our formulation is based on a spectral formulation of the square-root of $\mathbf{B}$, which we believe has not been used in any such inverse modelling system before. It allows horizontal and vertical error correlations of the initial concentration,

and horizontal and temporal error correlations of the flux to be represented. We provide full mathematical details. Our scheme allows the various correlation components to be switched on/off and for the respective length and timescales to be set in a way that is much more computationally efficient than representing such a $\mathbf{B}$-matrix explicitly.

   We test 14 configurations of the $\mathbf{B}$-matrix (including the diagonal configuration) in a 100 day test assimilation of surface flask measurements of methane. We measure the performance of each by comparing the analysis to unassimilated observations

held back for evaluation purposes. We find that the diagonal configuration is amongst the poorest performing choices of $\mathbf{B}$. The best performing choice uses the spectral method. It does not include correlations for the initial concentration field, but does account for spatio-temporal correlations for the fluxes. These have the form of a SOAR (second order auto-regressive) function with a correlation length-scale of $600\,\mathrm{km}$ and a timescale of 3 months. Our results demonstrate the effectiveness of our method, which is applicable to very high resolution inverse modelling systems. We propose that potential biases in the

prior initial condition field may be the reason for the poor performance when correlations in the prior initial concentration field are used.



## 1 Introduction

Top-down methods to estimate regional fluxes of atmospheric trace gases at the Earth's surface have been in use since the 1990s (Houweling et al., 1999). Such methods combine measurements of a trace gas from in-situ and/or remote sensing instruments
with information from chemical transport models to infer the surface fluxes. Trace gases that have been studied in this way include carbon dioxide (Feng et al., 2009; Chandra et al., 2022), carbon monoxide (Jiang et al., 2013; Zheng et al., 2019), methane ($CH_4$) (Fraser et al., 2013; Maasakkers et al., 2021; Qu et al., 2021; Wilson et al., 2021), nitrous oxide (Hirsch et al., 2006; Thompson et al., 2019), chlorofluorocarbons (An et al., 2012; Rigby et al., 2008), carbonyl sulfide (Ma et al., 2021), and volatile organic compounds like isoprene (Palmer et al., 2006), and formaldehyde (Gonzi et al., 2011). Knowledge of the
geographical distribution and magnitude of the fluxes of these gases is essential to understand the origin of pollutants that are damaging to the environment and/or health, and the natural cycle of compounds within the Earth system such as in the carbon cycle.

There is a range of methods used to make inroads into this problem, some of which are Bayesian inverse methods. The variational method (4DVar) (Chevallier et al., 2005) finds the mode of an assumed posterior distribution, which represents the
least squares fitting of a surface flux field to a-priori information and to observations of the trace gas. 4DVar assumes that prior statistics are Gaussian, with covariances that have been prescribed, but is otherwise an efficient and practical method. Other commonly used methods include the ensemble Kalman filter (EnKF) (Feng et al., 2009), which avoids the need to prescribe prior error covariances, but relies on a large enough and well constructed ensemble to produce reasonable results; and direct inversions (Gurney et al., 2002; McNorton et al., 2018; Chandra et al., 2022), which use an explicit formula to calculate directly
the posterior fluxes, but are restricted on computation grounds to a relatively small number of large-area surface regions. Other methods include the flixed-lag Kalman smoother (Zhuravlev et al., 2013), Markov chain Monte-Carlo (Lunt et al., 2019), regressions (Palmer et al., 2003), and back-trajectories (An et al., 2012).

It is possible to categorise these methods into those that (i) represent the fluxes on a grid that is at the same resolution as the underlying chemical transport model (Wilson et al., 2021), and (ii) represent only the aggregated fluxes over a relatively
small number of large-area regions on the globe (Locatelli et al., 2013). Contemporary 4DVar and EnKF methods tend to fall into category (i). These methods can infer surface fluxes with a high signal-to-noise ratio when the results are aggregated to large scales, but this ratio degrades when the scale is reduced to continental scale and smaller (for instance to over 150% of the prior at grid level (Locatelli et al., 2013; Chandra et al., 2022)). That said, there can still be significant variations in flux over continents. We therefore have a preference to study methods that are capable of resolving the fluxes at the model grid level.
This paper concerns developments to a leading 4DVar inversion method, namely INVICAT (Wilson et al., 2014), which is the 4DVar framework for the TOMCAT chemical transport model (Chipperfield, 2006).

One of the difficulties associated with using 4DVar is that the prior error covariances of the surface flux and initial condition fields need to be specified. This information is encapsulated in the **B**-matrix (see Sects. 2.2 and 3). The specification can include grid level variances, horizontal length-scales (potentially separately for land and sea, and dependent on location and
source type), vertical length-scales (for the initial conditions only), and temporal timescales (for the fluxes only). A first





challenge, given the large size of the **B**-matrix, is to just represent the correlation-related parts of **B** (related to the off-diagonal elements) in a practical way, and a second problem is to determine realistic values of the above quantities. With respect to the first problem, many studies revert to either a diagonal prior error covariance matrix (Mendoza-Dominguez and Russell, 2001; Wilson et al., 2014; Jiang et al., 2013), which requires the specification of the variances only, or to a partial representation of the correlations (e.g. omitting temporal correlations in the flux, Locatelli et al. (2013)). Off-diagonal elements in **B** (see e.g. Bannister (2008a)) are important in an inverse problem given that the prior fields for the flux and the initial condition field are each likely to have correlated errors (spatially and temporally), and so omitting the correlations would lead to a sub-optimal posterior (Houweling et al., 1999).

Given the large size of the **B**-matrix ($\sim \mathcal{O}(\text{state size}^2)$), the task of representing the correlations explicitly (e.g. to form the **B**-matrix and then compute $\mathbf{B}^{1/2}$ as a preconditioner) requires a large computational burden, which becomes infeasible with increasing model resolution. An alternative approach is the use of a more efficient method, which avoids explicitly representing the **B**-matrix. In this paper we describe and test a spectral method of efficiently modelling the **B**-matrix, largely adapted from operational data assimilation systems used for weather forecasting.

The structure of this paper is as follows. In Sect. 2 the INVICAT system is described and the basic problem of modelling **B** is outlined. In Sect. 3 INVICAT's original (diagonal) **B**-matrix is described and the spectral method is outlined. In Sect. 4 the specific prior error covariances are shown. In Sect. 5 the experiments that test the spectral method are detailed and in Sect. 6 the results are shown. Section 7 concludes the paper. Appendices A, B, and C give full mathematical details, and Appendix D compares the cost of the spectral scheme to that of representing the **B**-matrix explicitly.

## 2 The INVICAT 4DVar system and the prior error covariance matrix (B)

### 2.1 The INVICAT state and cost function

The state vector for the INVICAT system (Wilson et al., 2014), $\mathbf{x}$, is an augmentation of two kinds of trace gas field: (i) initial conditions, $\mathbf{c}^0$, which is a flattened form of the 3D field (a function of longitude, $\lambda$, latitude, $\phi$, and height, $z$) valid at the start of a 4DVar window, and (ii) the surface fluxes, $\boldsymbol{\rho}^0, \boldsymbol{\rho}^1, \dots \boldsymbol{\rho}^T$, which are flattened forms of the 2D fluxes, each a function of $\lambda$ and $\phi$, at a set of time steps within the window. In our case the time steps are separated by one month. The state vector may therefore be written as

$$\mathbf{x} = \begin{pmatrix} \mathbf{c}^0 \\ \hline \boldsymbol{\rho}^0 \\ \boldsymbol{\rho}^1 \\ \vdots \\ \boldsymbol{\rho}^T \end{pmatrix}. \tag{1}$$

There are special values of $\mathbf{x}$, namely the prior $\mathbf{x}^b$ (formally the mean of the prior distribution, also called the background state) and the posterior $\mathbf{x}^a$ (formally the mode of the posterior distribution, also called the analysis state). The INVICAT cost





function is

$$J_{\mathbf{x}}(\mathbf{x}) = \frac{1}{2}\left(\mathbf{x} - \mathbf{x}^{\mathrm{b}}\right)^{\mathsf{T}} \mathbf{B}^{-1}\left(\mathbf{x} - \mathbf{x}^{\mathrm{b}}\right) + \frac{1}{2}\left(\mathbf{y} - \mathbf{H}\mathbf{x}\right)^{\mathsf{T}} \mathbf{R}^{-1}\left(\mathbf{y} - \mathbf{H}\mathbf{x}\right), \tag{2}$$

where $\mathbf{B}$ is the background error covariance matrix (see below), $\mathbf{y}$ is vector of observation values spread throughout an assimilation window, $\mathbf{H}$ is the observation operator, and $\mathbf{R}$ is the observation error covariance matrix. $\mathbf{H}$ is a matrix operator that acts on a given $\mathbf{x}$ and outputs the model equivalents of the observations. Part of $\mathbf{H}$ represents the action of the TOMCAT model from $t = 0$ to each observation time within the window ($0 \le t \le T$) accounting for the $\mathbf{c}^0$ and $\boldsymbol{\rho}^t$ influences on the

trajectory, and part of $\mathbf{H}$ represents the interpolation of the fields at each observation time to the locations of the observations. $\mathbf{H}$ is therefore a complicated operator, given in a simplified form here for brevity.

### 2.2 Framework to representing the prior error covariance matrix

Minimising a problem of the form of Eq. (2) with respect to $\mathbf{x}$ is normally numerically badly conditioned. The condition number (see e.g. Sect. 11.4 of Lewis et al. (2006)), quantified as the ratio of the maximum-to-minimum eigenvalues of the

Hessian matrix, $\mathbf{B}^{-1} + \mathbf{H}^{\mathsf{T}}\mathbf{R}^{-1}\mathbf{H}$, is often large, owing partly to the typically high condition number of $\mathbf{B}$ (e.g. Gauthier et al. (1999)). Instead, minimisation is done with a control variable, denoted $\boldsymbol{\chi}$. Let

$$\delta\mathbf{x} = \mathbf{x} - \mathbf{x}^{\mathrm{b}} = \mathbf{B}^{1/2}\boldsymbol{\chi}, \tag{3}$$

where $\mathbf{B}^{1/2}$ is the control variable transform (CVT) and $\delta\mathbf{x}$ is an increment with respect to $\mathbf{x}^{\mathrm{b}}$. Equation (3) transforms (2) into a function of $\boldsymbol{\chi}$:

$$J_{\boldsymbol{\chi}}(\boldsymbol{\chi}) = \frac{1}{2}\boldsymbol{\chi}^{\mathsf{T}}\boldsymbol{\chi} + \frac{1}{2}\left(\mathbf{y} - \mathbf{H}\left[\mathbf{x}^{\mathrm{b}} + \mathbf{B}^{1/2}\boldsymbol{\chi}\right]\right)^{\mathsf{T}} \mathbf{R}^{-1}\left(\mathbf{y} - \mathbf{H}\left[\mathbf{x}^{\mathrm{b}} + \mathbf{B}^{1/2}\boldsymbol{\chi}\right]\right). \tag{4}$$

The cost function $J_{\boldsymbol{\chi}}(\boldsymbol{\chi})$ is typically better conditioned than $J_{\mathbf{x}}(\mathbf{x})$ because the background error covariance matrix in $\boldsymbol{\chi}$-space is the identity matrix. The gradient of $J_{\boldsymbol{\chi}}$ with respect to $\boldsymbol{\chi}$ is required for the descent algorithm:

$$\nabla_{\boldsymbol{\chi}}J_{\boldsymbol{\chi}} = \boldsymbol{\chi} - \mathbf{B}^{\mathsf{T}/2}\mathbf{H}^{\mathsf{T}}\mathbf{R}^{-1}\left(\mathbf{y} - \mathbf{H}\left[\mathbf{x}^{\mathrm{b}} + \mathbf{B}^{1/2}\boldsymbol{\chi}\right]\right). \tag{5}$$

Minimising $J_{\boldsymbol{\chi}}(\boldsymbol{\chi})$ in (4) is equivalent to minimising $J_{\mathbf{x}}(\mathbf{x})$ in (2) with $\mathbf{B} = \mathbf{B}^{1/2}\mathbf{B}^{\mathsf{T}/2}$ (where $\mathbf{B}^{\mathsf{T}/2} = \left(\mathbf{B}^{1/2}\right)^{\mathsf{T}}$). The minimum

of $J_{\boldsymbol{\chi}}$ is at $\boldsymbol{\chi}^{\mathrm{a}}$ and corresponds to the analysis state, i.e. $\mathbf{x}^{\mathrm{a}} = \mathbf{x}^{\mathrm{b}} + \mathbf{B}^{1/2}\boldsymbol{\chi}^{\mathrm{a}}$.

There are two categories of CVT discussed in this paper.

1. The first is when $\mathbf{B}^{1/2}$ is set to a diagonal operator (called $\mathbf{B}_{\mathrm{d}}^{1/2}$) where the diagonal elements are background error standard deviations. More information is given in Sect. 3.1.

2. The second gives an implied $\mathbf{B}$-matrix. It is sometimes possible to find $\mathbf{B}^{1/2}$ by first forming $\mathbf{B}$ and then decomposing

using an eigen- or Choslesky decomposition (see e.g. Sect. 9.1 of Lewis et al. (2006)). This is the procedure used in many 4DVar-based systems such as Chevallier et al. (2007) (Chevallier, personal communication). Here though it is assumed it is difficult or impossible to know $\mathbf{B}$ explicitly, so instead a plausible form of $\mathbf{B}^{1/2}$ (we call $\mathbf{B}_{\mathrm{sp}}^{1/2}$) is proposed




without forming $\mathbf{B}$ first. In high-dimensional systems, $\mathbf{B}$ can be too large to store explicitly, so $\mathbf{B}_{\mathrm{sp}}^{1/2}$ is coded as a set of computationally feasible steps that are designed to produce sensible background errors, $\mathbf{x} - \mathbf{x}^{\mathrm{b}}$, via (3) using control variables $\boldsymbol{\chi}$ which are uncorrelated and have unit background error covariance. There are numerous examples of $\mathbf{B}_{\mathrm{sp}}^{1/2}$ in the numerical weather prediction literature. For reviews of different applications and methods, see Bannister (2008b, 2017) and Bannister et al. (2020). Such an implied $\mathbf{B}$-matrix, $\mathbf{B}_{\mathrm{sp}}^{1/2}\mathbf{B}_{\mathrm{sp}}^{\mathsf{T}/2}$ can be orders or magnitude more efficient to use than an explicit matrix, but is approximate. More information is given in Sect. 3.2.

Both categories use the same cost function form (4) and only the form of $\mathbf{B}^{1/2}$ differs between the two.

## 3    Specific approaches used to model B

In this work we consider different configurations of $\mathbf{B}^{1/2}$ which includes (or not) spatial and temporal correlations with different correlation scales, in categories 1 or 2 above. These are described in this section, together with their expected advantages and disadvantages. The INVICAT system is described on a global grid with $n_x$ longitudes, $n_y$ latitudes, $n_z$ levels, and $T + 1$ flux times. The model grid is spaced regularly in the longitudinal direction and has irregular Gaussian latitudes. The vertical grid is based on a combination of terrain-following and pressure coordinates ($\sigma$-p) up to 0.1 hPa (see Wilson et al. (2014) and Chipperfield (2006) for more details). The experiments done in the paper have $n_x = 64$, $n_y = 32$, $n_z = 60$, and $T = 3$. This produces a grid cell size of approximately $5.6° \times 5.6°$. This is a relatively low resolution, chosen for illustration.

### 3.1    Diagonal B-matrix ($\mathbf{B_d}$)

A diagonal $\mathbf{B}$-matrix (labelled $\mathbf{B_d}$) is the simplest representation of the prior error distribution. It assumes that background errors are uncorrelated in space and time on the model grid. This is sub-optimal because true prior errors are likely to be correlated and also because the grid box sizes change with latitude. With $\mathbf{B_d}$, though, only the variances need to be specified. The control space for the for this choice (the space that $\mathbf{B_d}^{1/2}$ acts on), is denoted $\boldsymbol{\chi}_{\mathrm{d}}$, and has the same structure as (1) so there is a one-to-one correspondence between elements in $\delta\mathbf{x}$ and in $\delta\boldsymbol{\chi}_{\mathrm{d}}$.

Despite this simplicity, it can still be difficult to determine the variances well (to reflect the 'true' variances), since these are likely to depend on location and time. In this work, it is assumed that the background error standard deviations ($\sigma_c^{\mathrm{b}}(\lambda, \phi, z)$ and $\sigma_{\rho^t}^{\mathrm{b}}(\lambda, \phi)$, the square-roots of the variances of the initial conditions and fluxes respectively) are set to be a fraction of the a-priori field. Namely, $\sigma_c^{\mathrm{b}}(\lambda, \phi, z) = f_c c^{0\mathrm{b}}(\lambda, \phi, z)$, where $f_c = 0.1$ and $c^{0\mathrm{b}}(\lambda, \phi, z)$ is the background initial condition field; and $\sigma_{\rho^t}^{\mathrm{b}}(\lambda, \phi) = f_\rho \left|\rho^{t\mathrm{b}}(\lambda, \phi)\right|$, where $f_\rho = 0.4$ and $\rho^{t\mathrm{b}}(\lambda, \phi)$ is the background flux field[1]. Let the square-root of $\mathbf{B_d}$ have the

---

[1] Values of $\sigma_{\rho^t}^{\mathrm{b}}(\lambda, \phi)$ smaller than $10^{10}$ molecules cm$^{-2}$ s$^{-1}$ are replaced by $10^{10}$ molecules cm$^{-2}$ s$^{-1}$, providing a minimum flux error value.





following block form:

$$
\mathbf{B}_{\mathrm{d}}^{1/2} =
\begin{pmatrix}
\boldsymbol{\Sigma}_c^{\mathrm{b}} & 0 & 0 & 0 & 0 \\
0 & \boldsymbol{\Sigma}_{\rho^0}^{\mathrm{b}} & 0 & 0 & 0 \\
0 & 0 & \boldsymbol{\Sigma}_{\rho^1}^{\mathrm{b}} & 0 & 0 \\
0 & 0 & 0 & \ddots & 0 \\
0 & 0 & 0 & 0 & \boldsymbol{\Sigma}_{\rho^T}^{\mathrm{b}}
\end{pmatrix},
\tag{6}
$$

where $\boldsymbol{\Sigma}_c^{\mathrm{b}}$ is the diagonal matrix of background error standard deviations in concentration, $\sigma_c^{\mathrm{b}}(\lambda,\phi,z)$ (with $n_x n_y n_z$ diagonal elements) and $\boldsymbol{\Sigma}_{\rho^t}^{\mathrm{b}}$ is the diagonal matrix of background error standard deviations in flux at time $t$, $\sigma_{\rho^t}^{\mathrm{b}}(\lambda,\phi)$ (with $n_x n_y$ diagonal elements at each time).

The particular values of $f_c$ and $f_\rho$ mentioned above have been chosen by trial-and-error (not shown) in order to yield reasonably small mean deviations and root-mean-square deviations between the resulting analyses (using a trial period of 100-days) and some unassimilated observations of surface methane (see Sect. 5). The fact that $f_c < f_\rho$ means that the analysed surface fluxes are allowed to deviate more from the background than the initial conditions are (relative to the size their background values). The problem with setting the error standard deviations to be proportional to the background is that assimilated observations will struggle to update elements of the state vector with small background values. The procedure noted in footnote 1 will partially mitigate this effect, but the use of $f_c$ and $f_\rho$ remains a potential drawback of this approach. This option falls into category 1 in Sect. 2.2.

## 3.2 Spectrally modelled B-matrix ($\mathbf{B}_{\mathrm{sp}}$)

The main focus of this paper is the development of an efficient way to model a non-diagonal $\mathbf{B}$-matrix using the spectral method, here called $\mathbf{B}_{\mathrm{sp}}$. This is an example of category 2 in Sect. 2.2, where the $\mathbf{B}$-matrix is implied by the form of the CVT. The proposed square-root is as follows:

$$
\mathbf{B}_{\mathrm{sp}}^{1/2} =
\begin{pmatrix}
\boldsymbol{\Sigma}_c^{\mathrm{b}}\boldsymbol{\Xi}^{-1}\mathbf{F}_{\mathrm{v}c}\boldsymbol{\Lambda}_{\mathrm{v}c}^{1/2}\mathbf{F}_{\mathrm{v}c}^{\mathsf{T}}\mathbf{R}_{\mathrm{h}}\mathbf{S}_{\mathrm{h}}\boldsymbol{\Lambda}_{\mathrm{h}c}^{1/2} & 0 \\
0 & \boldsymbol{\Sigma}_\rho^{\mathrm{b}}\mathbf{F}_{\mathrm{t}\rho}\boldsymbol{\Lambda}_{\mathrm{t}\rho}^{1/2}\mathbf{F}_{\mathrm{t}\rho}^{\mathsf{T}}\mathbf{R}_{\mathrm{h}}\mathbf{S}_{\mathrm{h}}\boldsymbol{\Lambda}_{\mathrm{h}\rho}^{1/2}
\end{pmatrix},
\tag{7}
$$

which acts on a control variable of the form $\boldsymbol{\chi}_{\mathrm{sp}} = \left( \boldsymbol{\chi}_{c^0} \mid \boldsymbol{\chi}_\rho \right)^{\mathsf{T}}$. Evidently, this form CVT is more complicated than the previous form, so it deserves a step-by-step explanation. The definitions of the symbols that appear in (7) are described below, but we first make two remarks on the form of the control space. Firstly, like for $\mathbf{B}_{\mathrm{d}}^{1/2}$, the $c$ and $\rho$ parts in $\mathbf{B}_{\mathrm{sp}}^{1/2}$ are separate.





160 This means that $\mathbf{B}_{\mathrm{sp}}$ still assumes that errors between these two parts of the state vector are uncorrelated. Including these correlations is beyond the scope of this work. Secondly, the part of $\mathbf{B}_{\mathrm{sp}}^{1/2}$ that concerns $\rho$ couples different temporal components of the flux field, and so the $\rho$ part of the CVT cannot be written in a block form similar to (6). In other words, the parts of the CVT associated with $\rho$ include operators for all times together, and are no longer separate for each time. The parts of $\mathbf{B}_{\mathrm{sp}}^{1/2}$ associated with $c$ and $\rho$ are now explained in turn.

165    The first part of $\mathbf{B}_{\mathrm{sp}}^{1/2}$ is that associated with the initial conditions, $\mathbf{c}^0$ (upper left in (7)).

1. In the order of operation, the first stage, $\mathbf{S}_{\mathrm{h}}\mathbf{\Lambda}_{\mathrm{h}c}^{1/2}$, models horizontal correlations. This has a similar form to the square-root of an eigenvalue decomposition, where $\mathbf{\Lambda}_{\mathrm{h}c}$ is the matrix of eigenvalues, and $\mathbf{S}_{\mathrm{h}}$ is the matrix of eigenvectors. $\mathbf{S}_{\mathrm{h}}$ here is actually a spherical spectral transform (from a spectral space representation of the control vector to grid space), so the eigenvectors have the form of spherical harmonics. The form of the transform is symbolic only, as practically the horizontal transform is applied separately for each vertical level. The diagonals of $\mathbf{\Lambda}_{\mathrm{h}c}$ form prescribed functions of total wavenumber (0 to $L$, where $L$ is the maximum total wavenumber) and there is one function (called a horizontal variance spectrum) for each vertical level. For simplicity, we use the same function for each vertical level, which is equivalent to using a constant horizontal correlation length-scale for prior errors in $c$. $\mathbf{S}_{\mathrm{h}}$ uses the *SHTools* software Wieczorek et al. (2018). Notes on this part of the transform are given in Appendices A and B1, how $\mathbf{\Lambda}_{\mathrm{h}c}$ is found is described in Appendix C1, and an example spectrum is shown in Sect. 4.2.

2. The next stage is the horizontal reconfiguration operator, $\mathbf{R}_{\mathrm{h}}$ (do not confuse with $\mathbf{R}$ in the cost function), which linearly interpolates from the horizontal grid used by *SHTools* (a set of longitudes and Gaussian co-latitudes) to that used by INVICAT (a different set of longitudes and Gaussian latitudes).

3. The next stage, $\mathbf{\Xi}^{-1}\mathbf{F}_{\mathrm{v}c}\mathbf{\Lambda}_{\mathrm{v}c}^{1/2}\mathbf{F}_{\mathrm{v}c}^{\mathsf{T}}$, models vertical correlations. The eigenvector and eigenvalue-like matrices ($\mathbf{F}_{\mathrm{v}c}$ and $\mathbf{\Lambda}_{\mathrm{v}c}$ respectively, each $n_z \times n_z$ matrices) are derived from proposed vertical correlation matrices. Again, this transform is symbolic only, as practically the vertical matrices are applied separately for each horizontal position. We derive $\mathbf{F}_{\mathrm{v}c}$ as the eigenvectors of a globally-averaged vertical correlation matrix for $CH_4$ but we allow $\mathbf{\Lambda}_{\mathrm{v}c}$ to vary with latitude only. Practical notes are given in Appendix B2, how $\mathbf{F}_{\mathrm{v}c}$ and $\mathbf{\Lambda}_{\mathrm{v}c}$ are found is described in Appendix C2, and the resulting vertical covariance matrix is shown in Sect. 4.3.

4. The matrix $\mathbf{\Xi}$ is an adjustment matrix. Because $\mathbf{F}_{\mathrm{v}c}$ and $\mathbf{\Lambda}_{\mathrm{v}c}$ are not exact eigenvectors and eigenvalues of the *local* vertical covariance (see Appendix C2 for details), $\mathbf{F}_{\mathrm{v}c}\mathbf{\Lambda}_{\mathrm{v}c}^{1/2}\mathbf{F}_{\mathrm{v}c}^{\mathsf{T}}$ is not the square-root of a strict correlation matrix. $\mathbf{\Xi}$ ensures that the diagonal elements of $\left(\mathbf{\Xi}^{-1}\mathbf{F}_{\mathrm{v}c}\mathbf{\Lambda}_{\mathrm{v}c}^{1/2}\mathbf{F}_{\mathrm{v}c}^{\mathsf{T}}\right)\left(\mathbf{\Xi}^{-1}\mathbf{F}_{\mathrm{v}c}\mathbf{\Lambda}_{\mathrm{v}c}^{1/2}\mathbf{F}_{\mathrm{v}c}^{\mathsf{T}}\right)^{\mathsf{T}}$ are all unity (see Appendix C2).

5. The last matrix, $\mathbf{\Sigma}_c^{\mathrm{b}}$, is the same as in $\mathbf{B}_{\mathrm{d}}$, and specifies the standard errors in $c$, see Sect. 4.1.

The second part of $\mathbf{B}_{\mathrm{spec}}^{1/2}$ is that associated with the surface fluxes, $\boldsymbol{\rho}$ (bottom right in Eq. (7)).

6. The first stage is $\mathbf{S}_{\mathrm{h}}\mathbf{\Lambda}_{\mathrm{h}\rho}^{1/2}$. This is similar to point 1 above and again the form is symbolic only, as practically the transform is applied separately to each of the $T+1$ times. The diagonals of $\mathbf{\Lambda}_{\mathrm{h}\rho}$ form prescribed functions of total wavenumber





components and time (components 0 to $L$). For simplicity though, we use the same function for each time, which is equivalent to using a constant correlation length-scale for surface flux errors. See Appendices A, B1, and C1, and Sect. 4.2.

7. The reconfiguration operator, $\mathbf{R}_{\mathrm{h}}$, is the same as in point 2 above.

8. The next stage is $\mathbf{F}_{\mathrm{t}\rho}\mathbf{\Lambda}_{\mathrm{t}\rho}^{1/2}\mathbf{F}_{\mathrm{t}\rho}^{\mathsf{T}}$, which models temporal correlations for the surface flux. The temporal eigenvectors, $\mathbf{F}_{\mathrm{t}\rho}$, and eigenvalues, $\mathbf{\Lambda}_{\mathrm{t}\rho}$, are derived from a prescribed $(T+1) \times (T+1)$ temporal correlation matrix with a specified correlation timescale. See Appendices B3 and C3, and Sect. 4.4.

9. The last matrix, $\mathbf{\Sigma}_{\rho}^{\mathrm{b}}$, is the same as in $\mathbf{B}_{\mathrm{d}}$. $\mathbf{\Sigma}_{\rho}^{\mathrm{b}}$ is related to the individual $\mathbf{\Sigma}_{\rho^t}^{\mathrm{b}}$ ($0 \leq t \leq T$, as used in (6)) in the following
$(T+1) \times (T+1)$ block matrix:

$$
\mathbf{\Sigma}_{\rho}^{\mathrm{b}} = \begin{pmatrix} \mathbf{\Sigma}_{\rho^0}^{\mathrm{b}} & & \\ & \ddots & \\ & & \mathbf{\Sigma}_{\rho^T}^{\mathrm{b}} \end{pmatrix}. \tag{8}
$$

The horizontal, vertical, and temporal transforms can be individually 'switched off' (meaning individually replaced with the identity), to investigate their impact.

Appendix D shows how efficient the spectral scheme is by comparing the computational cost with that of an explicit repre-
sentation of $\mathbf{B}$.

## 4   Prior error statistics

In this section we show example settings used in this work for representation of the $\mathbf{B}$-matrices.

### 4.1   Prior standard deviations

Figure 1 (right panels) are selected fields of background error standard deviations for the initial concentration ($\sigma_c^{\mathrm{b}}(\lambda, \phi, z =$
$4.8\,\mathrm{km})$, panel b) and flux ($\sigma_{\rho^{t=1\mathrm{month}}}^{\mathrm{b}}(\lambda, \phi)$, panel d). These standard deviations are a fraction of the corresponding prior fields, $\sigma_c^{\mathrm{b}}(\lambda, \phi, z) = f_c c^{0\mathrm{b}}(\lambda, \phi, z)$ and $\sigma_{\rho^t}^{\mathrm{b}}(\lambda, \phi) = f_\rho \left| \rho^{t\mathrm{b}}(\lambda, \phi) \right|$ (panels a and c respectively). As described in Sect. 3.1, $f_c = 0.1$ and $f_\rho = 0.4$ were chosen (but see footnote 1, concerning adjustments to the standard deviations for the flux, which explains why the spatial patterns in panels (c) and (d) are not identical, unlike panels (a) and (b)). The prior for the initial concentration varies over $200\,\mathrm{ppm}$ with larger values in the northern hemisphere (NH) than in the southern hemisphere (SH), since the
majority of $CH_4$ sources are in the NH. The prior for the flux has source 'hot spots' over S America, north east United States of America, S and E Europe, equatorial Africa, India, China, and Indonesia. Methane is emitted to the atmosphere from a range of anthropogenic and natural sources, and these hot spots are likely to be from a combination of fossil fuel-related activity, agriculture, waste, wetlands, fires and others. The fluxes over the oceans are small, and there is a weak sink over Antarctica. The same standard deviations are used in the diagonal (Sect. 3.1) and spectrally-modelled (Sect. 3.2) $\mathbf{B}$-matrices.





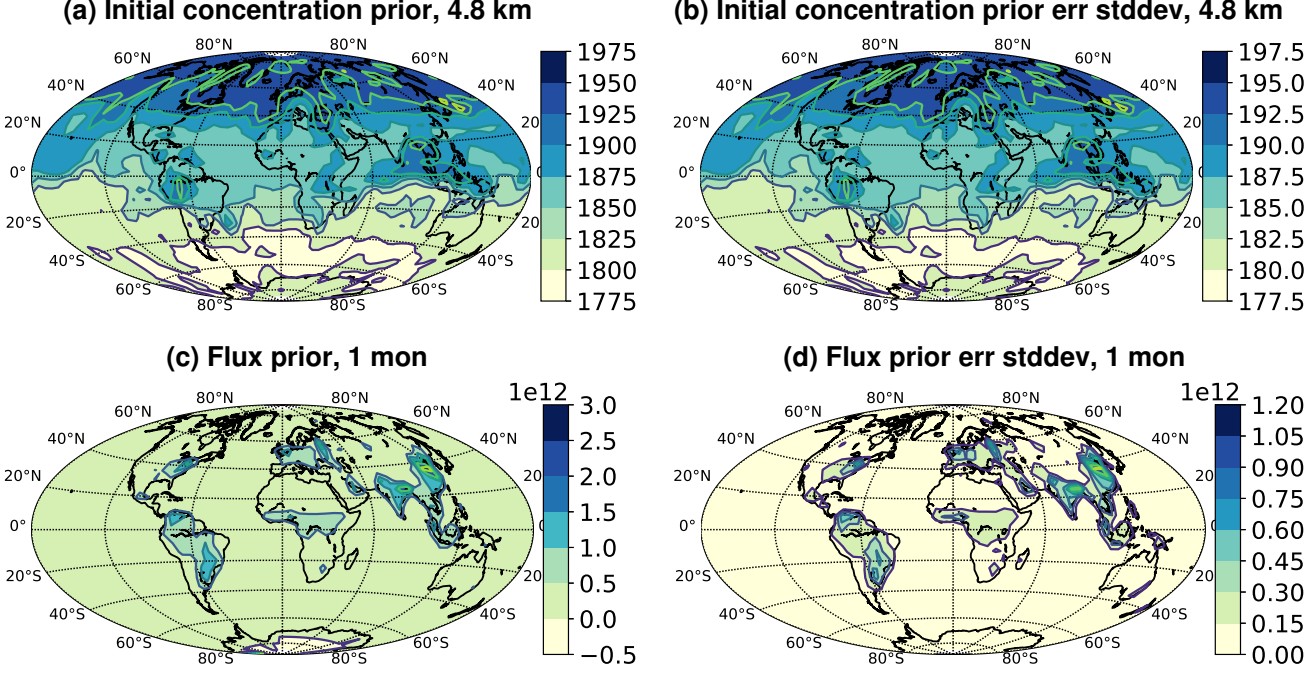

**Figure 1.** Panel (a) shows the prior initial concentration at altitude $4.8\,\mathrm{km}$ and panel (b) shows the prior initial concentration error standard deviation, both in parts-per-billion (ppb). Panel (c) shows the prior flux at $t = 1\,\mathrm{month}$ and panel (d) shows the prior flux error standard deviation, both in molecules $\mathrm{cm^{-2}\,s^{-1}}$.

### 4.2 Horizontal correlations of initial concentration and flux errors

The non-diagonal $\mathbf{B}$-matrix $\mathbf{B}_{\mathrm{sp}}$ uses correlation functions between any pair of points on the same horizontal level. These are specified in this paper as SOAR (second order auto-regressive) functions with specified length-scales. To illustrate, Fig. 2(a) shows two such SOAR functions, one with a length-scale of $400\,\mathrm{km}$ and another with $600\,\mathrm{km}$. These are example horizontal correlation functions for the initial concentration and flux respectively. For $\mathbf{B}_{\mathrm{sp}}$, these functions are used to produce the horizontal spectra $\mathbf{\Lambda}_{\mathrm{h}c}$ and $\mathbf{\Lambda}_{\mathrm{h}\rho}$ in Eq. (7) according to the procedure in Appendix C1. Although the formulation for $\mathbf{B}_{\mathrm{sp}}$ is constrained to produce only homogeneous and isotropic correlations, it results in a highly efficient CVT, which can be applied to high-resolution systems. The so-called 'variance spectra' for the two SOAR functions in Fig. 2(a) are shown in panel (b). Note that the spectrum for $\rho$ errors is narrower than that for $\mathbf{c}^0$ errors, which is the opposite of the correlation functions in real space. This is because the correlation function and spectra are related via the spectral transform (see Appendix C1), which, like the Fourier transform, treats distance and wavenumber as conjugate variable pairs.





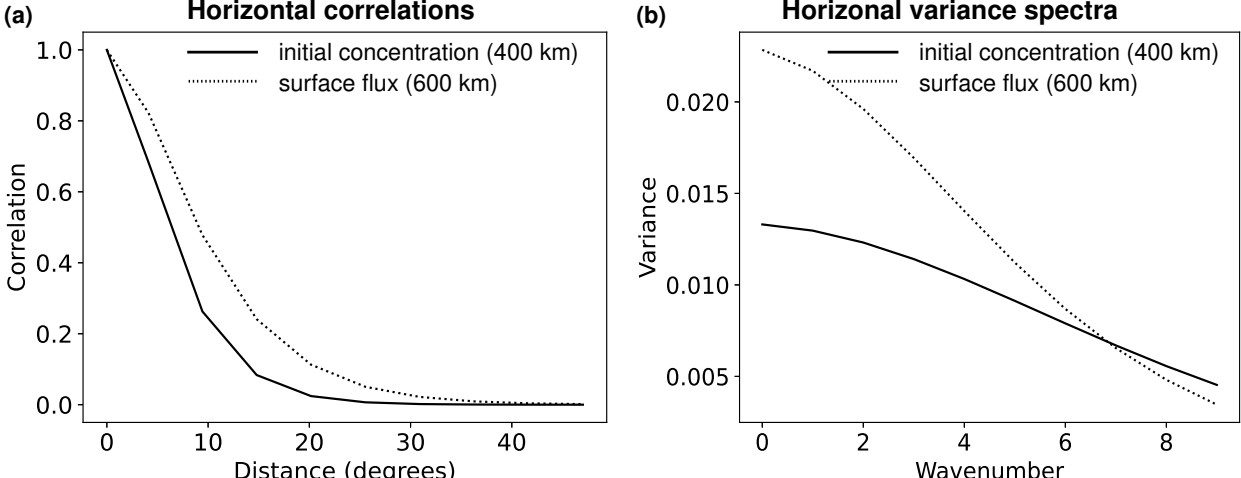

**Figure 2.** Panel (a) shows an example of two horizontal correlation functions of SOAR form with length-scales $400\,\mathrm{km}$ and $600\,\mathrm{km}$ (nominally applied to the initial concentration, $\mathbf{c}^0$, and flux fields, $\boldsymbol{\rho}^t$, respectively). Panel (b) shows their variance spectra (diagonal elements of $\boldsymbol{\Lambda}_{\mathrm{h}c}$ and $\boldsymbol{\Lambda}_{\mathrm{h}\rho}$ respectively). The procedure to compute the variance spectra is in Appendix C1.

### 4.3 Vertical correlations of initial concentration errors

$\mathbf{B}_{\mathrm{sp}}$ has the ability to include vertical covariances in $\mathbf{c}^0$ errors, which are described by the operators $\mathbf{F}_{\mathrm{v}c}$, $\boldsymbol{\Lambda}_{\mathrm{v}c}$, and $\boldsymbol{\Xi}^{-1}$. $\mathbf{F}_{\mathrm{v}c}$ contains the eigenvectors of the vertical correlation matrix, which is derived from a set of one-year methane forecasts made by INVICAT, valid between 1995 and 2004 (as described in Wilson et al. (2021)). This provides a time sequence of 10 January

$CH_4$ forecast fields, which are detrended and used to produce an estimate of the vertical covariance matrix mentioned above. This is shown in Fig. 3(a). The derived global eigenfunctions are used for vertical columns at every horizontal position, but the eigenvalue-like object, $\boldsymbol{\Lambda}_{\mathrm{v}c}$, is allowed to be latitude dependent. The remaining panels of Fig. 3 plot the vertical correlations $\left(\boldsymbol{\Xi}^{-1}\mathbf{F}_{\mathrm{v}c}\boldsymbol{\Lambda}_{\mathrm{v}c}^{1/2}\mathbf{F}_{\mathrm{v}c}^{\mathsf{T}}\right)\left(\boldsymbol{\Xi}^{-1}\mathbf{F}_{\mathrm{v}c}\boldsymbol{\Lambda}_{\mathrm{v}c}^{1/2}\mathbf{F}_{\mathrm{v}c}^{\mathsf{T}}\right)^{\mathsf{T}}$ for the model latitudes nearest $\sim 50°$ (b), $\sim 0°$ (c), and $\sim -50°$ (d), to show the range of vertical correlations that this is capable of representing. More details are given in Sect. 3.2 above and in Appendix C2.

### 4.4 Temporal correlations of flux errors

$\mathbf{B}_{\mathrm{sp}}$ has the ability to include temporal correlations in $\boldsymbol{\rho}$ errors, which are described by the operators $\mathbf{F}_{\mathrm{t}\rho}$ and $\boldsymbol{\Lambda}_{\mathrm{t}\rho}$. Figure 4(a) is an example correlation function of SOAR form with a timescale of one month. The corresponding correlation matrix has eigenfunctions $\mathbf{F}_{\mathrm{t}\rho}$ and eigenvalues $\boldsymbol{\Lambda}_{\mathrm{t}\rho}$, the latter are shown in panel (b) (there called a variance spectrum). The eigenvalues increase with eigenmode because of the way the eigensolver has ordered the modes.



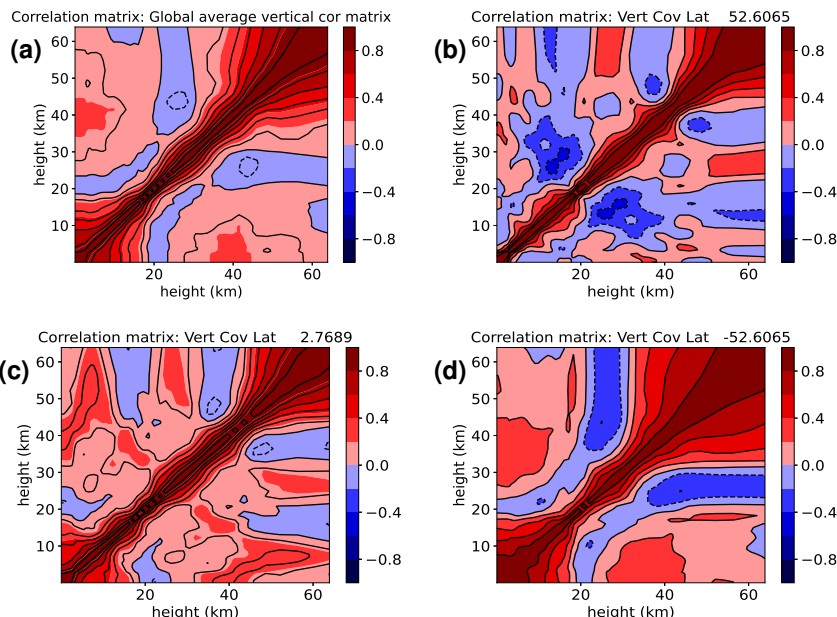

**Figure 3.** Selection of vertical background error correlation matrices for the initial concentration field, $\mathbf{c}^0$. Panel (a) is the global average correlation matrix, which is used to compute the global eigenvectors, $\mathbf{F}_{vc}$, and panels (b)-(d) are for implied correlation matrices for individual latitudes. Panel (b) $\sim 50°$, $(\mathbf{\Xi}^{50})^{-1}\mathbf{F}_{vc}\mathbf{\Lambda}^{50}_{vc}\mathbf{F}^{\intercal}_{vc}(\mathbf{\Xi}^{50})^{-1}$; (c) around the equator, $(\mathbf{\Xi}^{\mathrm{Eq}})^{-1}\mathbf{F}_{vc}\mathbf{\Lambda}^{\mathrm{Eq}}_{vc}\mathbf{F}^{\intercal}_{vc}(\mathbf{\Xi}^{\mathrm{Eq}})^{-1}$; and (d) $\sim -50°$, $(\mathbf{\Xi}^{-50})^{-1}\mathbf{F}_{vc}\mathbf{\Lambda}^{-50}_{vc}\mathbf{F}^{\intercal}_{vc}(\mathbf{\Xi}^{-50})^{-1}$. The eigenvectors are computed from the global mean vertical correlation matrix in (a), and the latitudinal dependent $\mathbf{\Lambda}^{\phi}_{vc}$ and $\mathbf{\Xi}^{\phi}$ matrices are computed according to the procedure in Appendix C2.

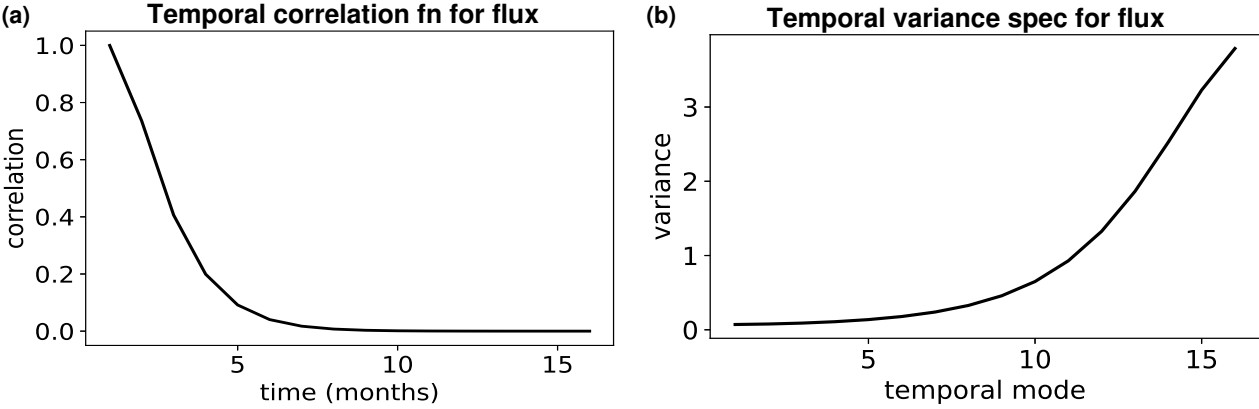

**Figure 4.** Panel (a) shows an example temporal correlation function for the flux fields, $\boldsymbol{\rho}^t$ (a SOAR function with a timescale of one month), and panel (b) shows its variance spectrum (diagonal elements of $\mathbf{\Lambda}_{t\rho}$). Note that the time axis in panel (a) refers to the time difference, so the temporal correlations apply forward and backward in time. The eigenvalue and eigenvector matrices ($\mathbf{\Lambda}_{t\rho}$ and $\mathbf{F}_{t\rho}$ respectively) are computed according to the procedure in Appendix C3.





### 4.5 Implied correlations of the combined horizontal–vertical (for c) and horizontal–temporal (for $\rho$)

Finally in this section, we put the above components together (apart from the standard deviations) to reveal the correlation structures implied by $\mathbf{B}_{\mathrm{sp}}$. The left column of Fig. 5 shows spatial error correlations for $\mathbf{c}^0$ associated with three impulse functions placed at the $11.4\,\mathrm{km}$ level (indicated by the three blue arrows in panel (c), see the caption for their locations). Panel (c) itself demonstrates the horizontal spreading, which has a homogeneous and isotropic SOAR length-scale of $400\,\mathrm{km}$. The vertical spreading of the impulse functions upwards and downwards are shown in (a-b) and (d-e) respectively. The right column of Fig. 5 shows spatio-temporal correlations for $\rho$ associated with three impulse functions placed at 3, 4, and 5 months (indicated by the three blue arrows in panels (g-i) respectively). The horizontal length-scale for $\rho$ in this example is longer than that of $\mathbf{c}^0$ ($600\,\mathrm{km}$), which is reflected in the broader horizontal functions. Each function is further spread in time, with the same timescale for each point (one month). A feature of these correlation structures is the series of small amplitude oscillations, which are evident where the correlation values are small (oscillations between the light yellow/green in all panels). These artifacts are a consequence of the spectral representation, and are thought to be related to the Gibbs phenomenon associated with Fourier transforms. They become more prominent when the correlations are narrow, and so are more evident in the left column. Since these are only of a small amplitude (maximum amplitude 0.001 in correlation), we assume that they will not affect the quality of the results (we shall see in Sect. 6.2 that they are negligible in typical analysis increment fields).

### 5 Experimental setup for assimilations

As explained in the introduction, the background error covariances have been implemented in the INVICAT system Wilson et al. (2014). For this paper, flux inversions are performed for observations over the first 100 days in 2018. This relatively short period of time allows the influence of the $CH_4$ initial conditions (as well as the surface fluxes) to be studied, and allows the problem to be studied without prohibitive cost. The observations assimilated are from 60 surface stations provided by the National Oceanic and Atmospheric Administration's Global Monitoring Laboratory (NOAA GML), making weekly or bi-weekly flask observations of methane (Fig. 6). $CH_4$ from the whole air samples inside the flasks is measured using gas chromatography with a flame ionisation detection method (Dlugokencky et al., 2018). The prior initial concentration is taken from a previously-run configuration INVICAT inversion with diagonal background error covariances, which was first initialised for 2009 and assimilated surface flask data, described in Wilson et al. (2021). The prior fluxes are taken from a range of bottom-up, satellite-based and model-based estimates, also described in Wilson et al. (2021). Emissions for the largest emission sectors, namely anthropogenic emissions, wetlands and biomass burning, are taken from the EDGAR v4.2 FT 2010 inventory (Olivier et al., 2012), the JULES land surface model (Clark et al., 2011), and the GFED v4.1 inventory (Van Der Werf et al., 2017) respectively. Other emissions, including the sink of $CH_4$ through methanotrophy in soils, are described in Wilson et al. (2021). The model takes meteorological data from the European Centre for Medium-Range Weather Forecasts' (ECMWF) ERA5 reanalyses (Hersbach et al., 2020).

This study assimilates real data, and as such there is no objective truth to evaluate against. Consequently, the observations are partitioned: 75% of the observations ($\sim 500$) are used for assimilation and the remaining 25% are used for evaluation.




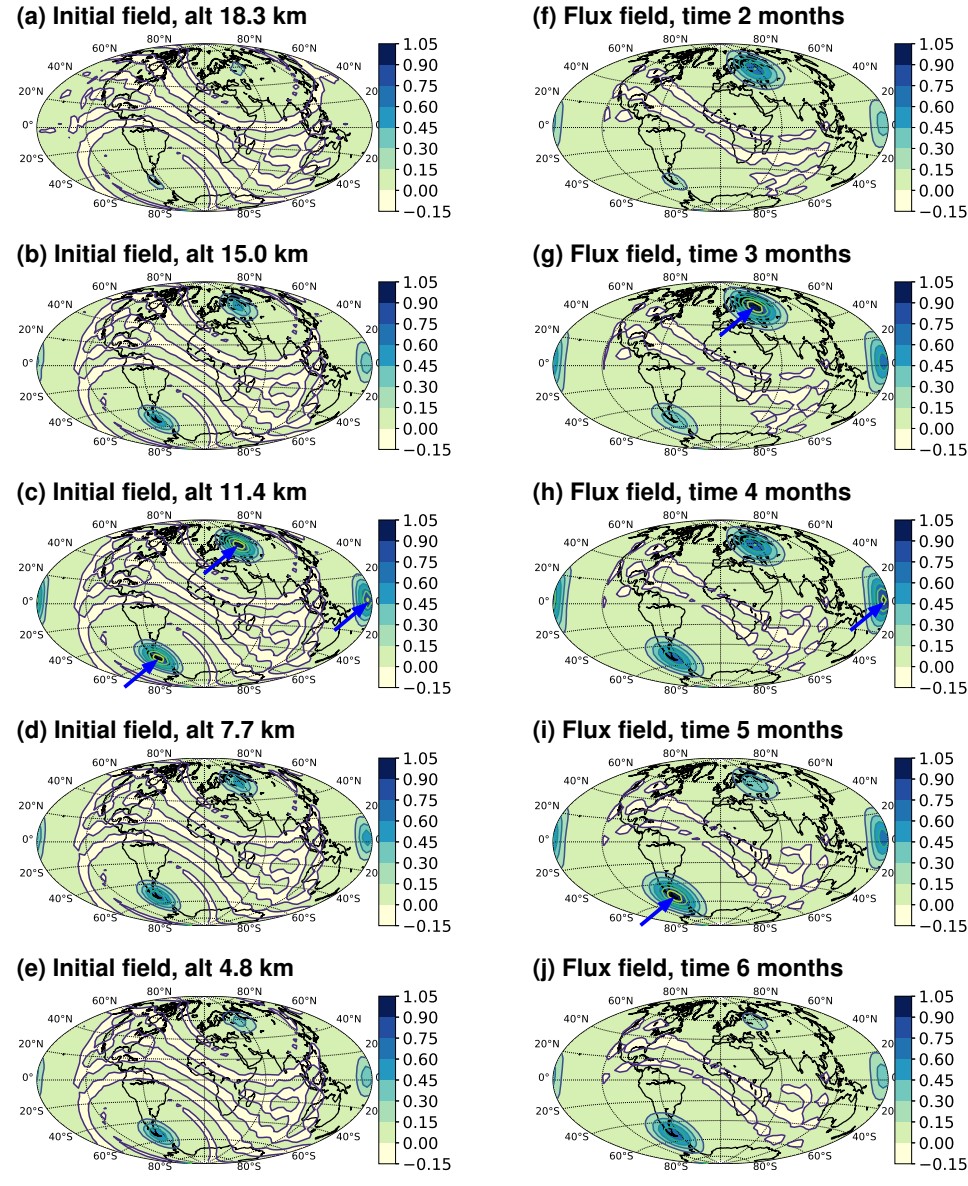

**Figure 5.** Implied correlations of the spectrally modelled **B**-matrix associated with a range of points (blue arrows). Panels (a-e) are long/lat structures at a range of heights for the initial concentration found from $\left(\boldsymbol{\Xi}^{-1}\mathbf{F}_{vc}\boldsymbol{\Lambda}_{vc}^{1/2}\mathbf{F}_{vc}^{\intercal}\mathbf{R}_h\mathbf{S}_h\boldsymbol{\Lambda}_{hc}^{1/2}\right)\left(\boldsymbol{\Xi}^{-1}\mathbf{F}_{vc}\boldsymbol{\Lambda}_{vc}^{1/2}\mathbf{F}_{vc}^{\intercal}\mathbf{R}_h\mathbf{S}_h\boldsymbol{\Lambda}_{hc}^{1/2}\right)^{\intercal}$ acting on a zero field apart from unit impulses at three points (lon,lat,alt): $(56°, 58°, 11.4\,\mathrm{km})$, $(174°, 3°, 11.4\,\mathrm{km})$, and $(293°, -47°, 11.4\,\mathrm{km})$. Panels (f-j) are long/lat structures at a range of times for the flux field found from $\left(\mathbf{F}_{t\rho}\boldsymbol{\Lambda}_{t\rho}^{1/2}\mathbf{F}_{t\rho}^{\intercal}\mathbf{R}_h\mathbf{S}_h\boldsymbol{\Lambda}_{h\rho}^{1/2}\right)\left(\mathbf{F}_{t\rho}\boldsymbol{\Lambda}_{t\rho}^{1/2}\mathbf{F}_{t\rho}^{\intercal}\mathbf{R}_h\mathbf{S}_h\boldsymbol{\Lambda}_{h\rho}^{1/2}\right)^{\intercal}$ acting on a zero field apart from unit impulses at three points (lon,lat,time): $(56°, 58°, 3\,\mathrm{months})$, $(174°, 3°, 4\,\mathrm{months})$, and $(293°, -47°, 5\,\mathrm{months})$. The correlation matrices shown are based on Eq. (7). The standard deviations $\boldsymbol{\Sigma}_c^{\mathrm{b}}$ and $\boldsymbol{\Sigma}_\rho^{\mathrm{b}}$ are omitted to show the implied correlations, rather than the covariances. The initial concentration has vertical correlations as Fig. 3 and a horizontal correlation length-scale of $400\,\mathrm{km}$ as Fig. 2 (continuous line). The flux has temporal correlations of one month as Fig. 4 and a horizontal correlation length-scale of $600\,\mathrm{km}$ as Fig. 2 (dotted line).





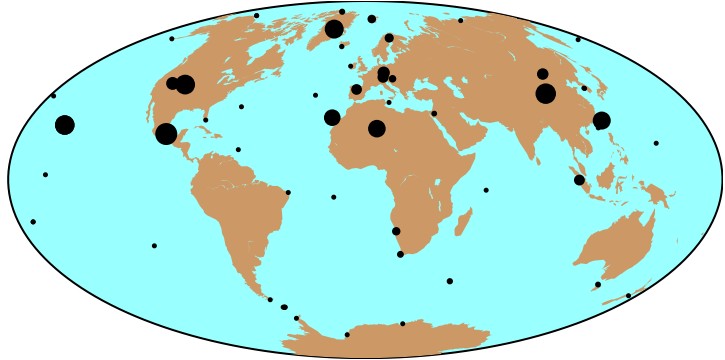

**Figure 6.** Map showing the locations of the 60 surface stations monitoring methane used in this study. These stations contribute to NOAA's global flask monitoring network. The size of each dot represents the altitude above sea level of the station, with the largest dot being $\sim 4.5\,\mathrm{km}$ in altitude. To avoid dots being too small to see, stations with an altitude $< 100\,\mathrm{m}$ are shown with a size corresponding to $100\,\mathrm{m}$ altitude.

Which observations are used for which purpose are decided at random but the same partition is used for tests with 14 different **B**-matrix configurations (see below). In order to test statistical significance of the results, we run this procedure a total of

five times, each with the same proportion of assimilation and evaluation observations, but using five sets of different random numbers to partition between assimilation or evaluation.

There are many parameters to control the specification of the **B**-matrix. Firstly, $f_c$ and $f_\rho$ control the background error standard deviations of the initial concentration and flux fields respectively. We assume that the values $f_c = 0.1$ and $f_\rho = 0.4$ are reasonable choices (Sect. 3.1), and so variations in these quantities are not explored. Of the eight variables that control

the correlations in **B**, there are five 'switch' variables. One is the choice of approach to modelling **B**: no correlations ($\mathbf{B}_\mathrm{d}$, Sect. 3.1) and the non-diagonal spectral approach ($\mathbf{B}_\mathrm{sp}$, Sect. 3.2). The other four are used to set whether or not to use vertical correlations and horizontal correlations for background errors in $\mathbf{c}^0$, and whether or not to use temporal correlations and horizontal correlations for background errors in $\boldsymbol{\rho}$. The remaining three variables concern the spatio-temporal correlation scales: the horizontal correlation length-scale for $\mathbf{c}^0$, the horizontal correlation length-scale for $\boldsymbol{\rho}$, and the temporal correlation

timescale for $\boldsymbol{\rho}$. A zero value in any of these latter three parameters is executed by switching off that particular correlation model. Running INVICAT is quite costly and so it is not possible to run with different settings as systematically as might be desired, but as mentioned above, we test with 14 configurations. These are listed in Table 1.



## 6 INVICAT performance with the B-matrices tested

### 6.1 Fit to evaluation observations

Table 1 lists the different configurations of the **B**-matrix tested. We evaluate the performance of each configuration by the reduction in the variance of the "observation-minus-model" statistic, $\kappa$, that is

$$\kappa = \left\langle \left( \mathbf{y}_{\mathrm{eval}} - \mathbf{H}_{\mathrm{eval}} \left[ \mathbf{x}^{\mathrm{a}} \right] \right)^2 \right\rangle - \left\langle \left( \mathbf{y}_{\mathrm{eval}} - \mathbf{H}_{\mathrm{eval}} \left[ \mathbf{x}^{\mathrm{b}} \right] \right)^2 \right\rangle, \tag{9}$$

where $\mathbf{y}_{\mathrm{eval}}$ is the vector of unassimilated evaluation observations, $\mathbf{H}_{\mathrm{eval}} \left[ \mathbf{x} \right]$ is the model equivalent (with $\mathbf{x} = \mathbf{x}^{\mathrm{a}}$ or $\mathbf{x}^{\mathrm{b}}$), and $\langle \bullet \rangle$ is the average over the evaluation observations. There are thus five different values of $\kappa$ for each **B**-matrix configuration –

one for each of the different sets of randomly chosen observation partitions.

The means and standard deviations of $\kappa$ (computed over the five values) are shown in Table 1 for each **B**-matrix. The configurations are ordered in descending order of the mean of $\kappa$, so the best results are towards the bottom of the table. The standard deviations of $\kappa$ give an indication of the significance that the means are different between experiments, but a more formal test is given in the last column. Considering the diagonal **B**-matrix as the control experiment, the level of significance

that the results with the alternative configurations are truly different from the control (rather than being by chance) are found from the Kolmogorov–Smirnov test (e.g. Press et al. (2007)). The higher the significance, the higher the chance that there is a meaningful impact in the configuration.

In the following, the configurations are referred to by their labels (A-N), given in brackets. It is evident from Table 1 that much improvement can be made on the diagonal configuration (B) as all but one of the configurations produce better

analyses when measured with the $\kappa$ statistic (bold column). The only configuration that is worse is (A), which uses vertical and horizontal correlations in $\mathbf{c}^0$, and horizontal and temporal correlations in $\boldsymbol{\rho}$. By looking at other rows in the table it becomes evident that this degradation is due to the correlations introduced in $\mathbf{c}^0$. When the horizontal correlations in $\mathbf{c}^0$ are removed the performance improves (C), which becomes better than (B), but with low significance. Additionally turning off the vertical correlations gives further improvements (I) with high significance. This and the remainder of the tests do not use vertical

correlations. Configurations (D, E, F, I) test different (decreasing) values of correlation length-scales of errors in $\mathbf{c}^0$, and produce progressively better results with shorter length-scales, culminating in the best results for this group being when these correlations are switched off, as mentioned above (I). The remaining tests in the table therefore do not use spatial correlations in $\mathbf{c}^0$ at all. Configurations (G,H,I,L,N) test different correlation timescales for $\boldsymbol{\rho}$ with better results for increasing timescales tested up to three months. Configurations (J,K,L,M) test different horizontal correlation length-scales in $\boldsymbol{\rho}$ and show there there

is little sensitivity to the specific value between 200 and 800 km.

### 6.2 Selection of analysis increments

Figure 7 plots a selection of analysis increments from configurations (A), (B), and (N), as described in Table 1 (the worst, control (diagonal **B**), and best respectively). The initial concentration increments (left column) are each added to the prior concentration given in Fig. 1(a), and the flux increments (right) are added to the prior flux given in Fig. 1(c) to give the



| Config. | B | Vert ($\mathbf{c}^0$) | Horiz ($\mathbf{c}^0$), km | Horiz ($\rho$), km | Temp ($\rho$), months | mean of $\kappa$ | stddev of $\kappa$ | Level of significance (%) difference from diagonal config |
|---------|---|---------|-----------|--------|--------|------------|------------|-------------------------|
| A | spectral | Y | 400. | 600. | 1.0 | **-68.77** | 48.30 | 30.26 |
| **B** | **diagonal** | **N** | **0.** | **0.** | **0.0** | **-82.29** | **12.82** | **00.00** |
| C | spectral | Y | 0. | 600. | 1.0 | **-86.38** | 37.94 | 30.26 |
| D | spectral | N | 600. | 600. | 1.0 | **-99.35** | 36.82 | 30.26 |
| E | spectral | N | 400. | 600. | 1.0 | **-100.41** | 31.91 | 30.26 |
| F | spectral | N | 200. | 600. | 1.0 | **-104.54** | 29.07 | 79.10 |
| G | spectral | N | 0. | 600. | 0.0 | **-104.57** | 11.40 | 96.39 |
| H | spectral | N | 0. | 600. | 0.5 | **-109.57** | 14.79 | 96.39 |
| I | spectral | N | 0. | 600. | 1.0 | **-114.25** | 18.68 | 99.62 |
| J | spectral | N | 0. | 200. | 2.0 | **-117.07** | 24.19 | 99.62 |
| K | spectral | N | 0. | 800. | 2.0 | **-118.22** | 22.28 | 99.62 |
| L | spectral | N | 0. | 600. | 2.0 | **-118.73** | 23.32 | 99.62 |
| M | spectral | N | 0. | 400. | 2.0 | **-118.98** | 24.39 | 99.62 |
| N | spectral | N | 0. | 600. | 3.0 | **-120.71** | 25.75 | 99.62 |

**Table 1.** Configurations of the 14 **B**-matrices tested and their performances with respect to unassimilated evaluation data. **Column 2** is the model used: *diagonal* (Sect. 3.1) and *spectral* (Sect. 3.2); **column 3** states whether vertical correlations are included in the **B**-matrix for the $\mathbf{c}^0$ field; **column 4** gives the horizontal correlation length-scale for $\mathbf{c}^0$; **column 5** gives the horizontal correlation length-scale for $\rho$; and **column 6** gives the temporal timescale for $\rho$. **Column 7** gives the means of $\kappa$ (Eq. (9)) over the five observation networks; and **column 8** gives the standard deviations. All experiments give negative $\kappa$ meaning that the analyses are closer to the evaluation observations than the backgrounds are. The more negative the result, the better the fit to the evaluation data. The table is ordered in descending value of mean $\kappa$ (bold column), so the best results are listed at the bottom of the table. **Column 9** gives the percentage level of statistical significance that the distribution of $\kappa$ values (over the five differing network experiments) differs from the distribution of $\kappa$ values over the control experiment (the diagonal configuration, the bold row). The closer the value of the significance to 100%, the smaller the chance that the results would have happened by chance.



respective posterior fields. Showing the increments allows us to study the experiments' differences more clearly than showing the analyses themselves.

Looking at the increments to the initial concentration, Fig. 1(a,c,e), (A) unsurprisingly has the largest values. This is because it is the only configuration shown that includes vertical correlations in $\mathbf{B}$, allowing surface innovations in the concentration to be spread directly to elevated levels. Configurations (B) and (N) do not include such direct correlations, and rely on effects associated with the propagation of covariances like $\mathbf{BM}_t^\mathsf{T}$ (where $\mathbf{M}_t$ is the model propagation from 0 to the time $t$ of an observation). Initial condition increments for (B) and (N), Fig. 1(c,e) are not dissimilar in magnitude and pattern because these two share the same configuration for the part of the $\mathbf{B}$-matrix concerning initial concentrations.

It is interesting to compare the locations of the surface stations in Fig. 6 with the initial concentration increments in the left panels of Fig. 7. Some of the peaks in Fig. 7(a) are directly above a measuring station, such as the one over the Northern Atlantic Ocean just East of North America (specifically Tudor Hill, Bermuda, station code BMW). Other increments are shifted, such as the negative one over Kazakhstan, which is a region with no local surface observations assimilated. This peak must therefore be a consequence of observation(s) made downstream via the $\mathbf{BM}_t^\mathsf{T}$ effect mentioned above. Interestingly, this and other increments are not present in (B) and (N), meaning that they are likely anomalous artifacts of potentially unrealistic vertical correlations.

Turning now to the surface fluxes, (A) and (N), Fig. 7 (b,f), have similar patterns and length-scales because (A) and (N) have the same length-scale settings for flux. The magnitude of the flux increments in (A) though are smaller than those in (N). Configuration (A) has a timescale of 1 month, while (N) has 3 months, so there is stronger influence of more observations (from different times) in (N). Configuration (B) has the smallest flux increments at most locations. It uses no temporal correlations, so there is no direct influence of observations made beyond 1 month from the start of the experiment. Looking at (N) as the best result, the flux increments have peaks over continental Europe, Eastern Europe, the Persian Gulf, and North Eastern China. Most of these locations do not coincide with observing stations, but they do all coincide with regions of high assumed a-priori uncertainty, see Fig.1(c). This illustrates that the inverse problem, in its best form (N), can update fluxes which are assumed most uncertain, even in the absence of nearby observations. The difference between panels (d) and (f) in particular highlight some important differences between the $\mathbf{B}$-matrices of the 'diagonal-$\mathbf{B}$' INVICAT and the one described in this paper.

## 6.3 Land, sea, and total fluxes

Figure 8 shows how the fluxes (aggregated over the globe, over land, and over sea points) change in time over the 100 days of the observation period (January to April 2018), and beyond to June. The prior and posteriors for the three configurations (A), (B), and (N) are shown. Looking at the total fluxes (blue and cyan lines in Fig. 8), the prior (blue) has the largest values, which dip slightly in February. The posterior for the diagonal configuration (B, dotted cyan line) also has this dip, but all values are lower by about 3%. The best configuration (N, dot-dashed cyan) is lower still, by about 7% from the prior. The worst performing configuration (A, continuous cyan line) has the lowest total fluxes, by about 9% from the prior. These results show that it is probable that the prior flux values are too high, but the 'wrong' configuration of the $\mathbf{B}$-matrix can reduce values too

**(a) Config. (A), initial concentration inc. 4.8 km**

**(b) Config. (A), flux inc. Jan 2018**

**(c) Config. (B), initial concentration inc. 4.8 km**

**(d) Config. (B), flux inc. Jan 2018**

**(e) Config. (N), initial concentration inc. 4.8 km**

**(f) Config. (N), flux inc. Jan 2018**

**Figure 7.** Analysis increments of the initial concentration at altitude $4.8\,\mathrm{km}$ (left column) and the surface flux at 1 month (right) for three configurations using the first observation subset considered. Configurations (A) (top row), (B) (middle), and (N) (bottom) are the worst, control (diagonal $\mathbf{B}$), and best performing configurations of the $\mathbf{B}$-matrix (see Table 1). Note that each map is plotted with a different scale, in order to to see the increments.





much. A similar pattern of results are found for the land-only points (red and orange lines), which comprise approximately 97.5% of the total fluxes in February.

The sea-only fluxes are much smaller (green lines), which comprise approximately 2.5% of the total fluxes in February. All posterior fluxes shown are still smaller than the prior, but the best configuration (N, dot-dashed light green line) this time has the smallest values rather than the worst configuration. All sea fluxes are broadly increasing at this time of the year.

Note that, apart from the diagonal configuration (B), all posterior flux amounts are different from the prior over all months shown, even though the observations go up to the start of April only. This is expected as configurations (A) and (N) include

temporal correlations. This does mean that persistent negative increments in the flux for the early (observed) part of the year infer negative increments later in the year. It will require much longer test runs to find out how useful such temporal correlations are. The total posterior flux values are comparable to other studies. Although we believe our evaluation approach is sufficient to test the different $\mathbf{B}$-matrices, the 3-month test period obviously makes it impossible to compare the seasonal cycle of $CH_4$ to that in other studies.

## 7    Conclusions


This paper documents our work to expand the capability of the $\mathbf{B}$-matrix in the 4DVar-based INVICAT system beyond the original diagonal INVICAT setup. This is done by efficiently allowing cross covariances in a-priori errors to be exploited. Cross covariances include horizontal and vertical correlations for the methane initial concentrations, and horizontal and temporal correlations for the surface methane flux. We model the correlations in the 4DVar as a control variable transform, which requires

only the 'square-roots' of the correlation matrices to be represented. As we use a spectral representation of the horizontal correlations, we call this a spectral method. We show how this method works in detail. To our knowledge this is the first time that such a method has been used for a flux estimation problem. The spectral method is very efficient. It is applicable to systems with very high resolutions, where existing methods that explicitly represent the $\mathbf{B}$-matrix would not be feasible (Appendix D), and to other chemical species.

The scheme allows us to switch on/off each of the above-mentioned correlation categories, and to choose the horizontal length-scales of a-priori errors in the initial concentration and flux, and the temporal timescale of errors in the flux. This allowed us to test out different $\mathbf{B}$-matrix configurations (including the standard diagonal setup) by systematic trial and error assimilations over a 100 day test period at the start of 2018. We assimilate surface observations from NOAA GML's flask measurement network. As it is virtually impossible to evaluate the analysed methane fluxes, we instead evaluate experiments

against subsets of unassimilated surface methane observations from the same network. This is repeated using five different randomly selected subsets to increase the robustness of the results. We find the mean-squared differences between the evaluation observations and the model's analysed equivalents, and then determine the performance of each $\mathbf{B}$-matrix configuration using the reduction of this quantity between the particular posterior and the prior. This is the $\kappa$ statistic in Eq. (9). As the experiments are relatively expensive to run we make a judicious choice of 14 configurations.



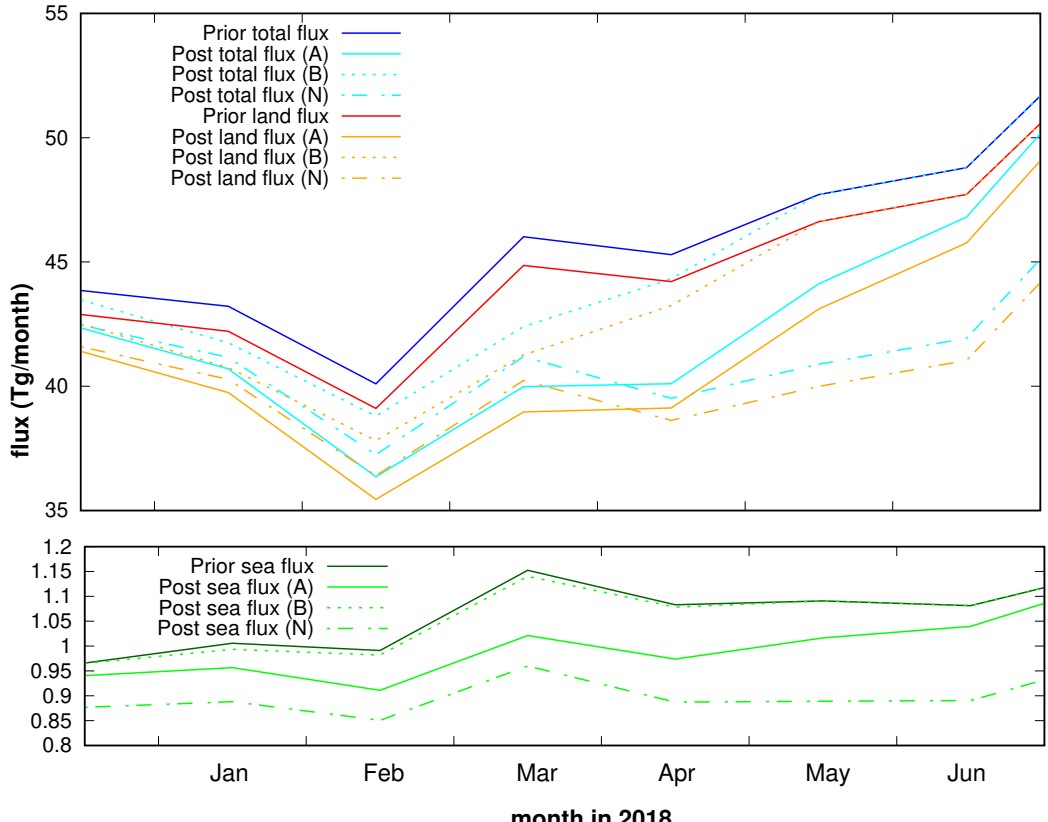

**Figure 8.** Time sequences of surface fluxes aggregated over the whole globe, over land points (both in top part) and over the sea (bottom part) for the first observation network considered. Prior and posterior values are shown for configurations (A), (B), and (N), which are the worst, control (diagonal **B**),and best performing configurations of the **B**-matrix (see Table 1

The configuration of the **B**-matrix has a significant impact on the results, which is an aspect that is usually ignored in most works. We find that considerable improvements can be made over the diagonal setup (configuration B). Introducing horizontal or vertical prior error correlations of the initial concentration is not an optimal configuration for the **B**-matrix in this system (configurations A and C). When used together, they can degrade the performance. The best configuration was found to have no spatial background error correlations for the initial concentration of methane, but a 600 km SOAR (second order auto-regressive) length-scale and 3 month SOAR timescale for the fluxes (configuration N). We have attempted to add robustness to the results by repeating the assimilation experiments with five different random divisions of the 665 observations between those that are used for assimilation ($\sim 75\%$) and those that are used for evaluation ($\sim 25\%$). Within the context of the experiments, this has allowed us to estimate the statistical significance of configuration N results as 'high'.

We acknowledge the limitations of this study. Partly for cost reasons it is restricted to one 100-day time period of observations. We would expect some variation of the results for different start times and run lengths, but we would not expect the



general conclusions to be different, although this should be tested. Longer run lengths would likely to be less sensitive to the configurations that describe errors in the initial concentration, which is an aspect that we are also interested in. We do not account for background error correlations between the initial concentration and the fluxes. We use a single lengthscale for each of the initital concentration and flux errors, while others use different lengthscales for land and ocean. For example Thanwerdas et al. (2022) use e-folding lengthscales of $500\,\text{km}$ over land and $1000\,\text{km}$ over sea for the flux, found presumably by forming $\mathbf{B}$ explicitly and then finding $\mathbf{B}^{1/2}$ by decomposition (see bullet point 2 in Sect. 2.2). Our spectral method assumes homogeneous error correlation structures (i.e. a single lengthscale), but land/sea differences could be accounted for by using two sets of spectral control variables (one set for the land and the other for the sea) and then using land/sea masks in model space. This procedure would also decouple the land and sea fluxes. This could also be extended to different source types. Like other studies, we use a single timescale, but this could easily be made dependent on the land/sea/source type. We do not evaluate the methane fluxes directly, but instead evaluate against methane observations, whose model equivalents are indirectly related to the fluxes when the air mass sampled was last influenced by surface processes. For this reason the observations later in the 100-day window are likely to be more valuable in this respect than those early in the window.

A remaining question is why does accounting for background error covariances for the initial concentrations at worst degrade the assimilation performance, and at best have little improvement over the diagonal $\mathbf{B}$ configuration (configurations A and C)? This is a pertinent question because we would expect errors in the a-priori methane concentration to have spatial correlations since this field is produced by a model that includes dynamical advection. If the initial concentration has a significant bias then any non-diagonal $\mathbf{B}$ configuration for the initial concentration would spread biased innovations to other locations and conceivably make the results worse than using a diagonal $\mathbf{B}$-matrix. Investigating this issue (e.g. by bias correction of the a-priori) is an avenue for further work. We hope that the spectral method will be adopted by other inverse modelling systems.

*Code availability.* Code and documentation are available via Bannister and Wilson (2024).

## Appendix A: The forward spectral transform

### A1 The forward spectral transform

The forward spectral transform, $\mathbf{S}_{\text{h}}$, changes the representation of a field increment from spectral space (a function of the total wavenumber, $l$, and zonal wavenumber, $m$, integers) to real space (a function of longitude, $\lambda_i$, and co-latitude, $\varphi_j$). Co-latitude is zero at the north pole, and so is related to latitude, $\phi$, via $\varphi = 90 - \phi$. The basis functions of the spectral representation are the spherical harmonic functions, $Y_{lm}(\theta, \varphi)$:

$$Y_{lm}(\theta, \varphi) = \begin{cases} \bar{P}_{lm}(\cos\varphi)\cos(m\theta) & m \geq 0 \\ \bar{P}_{l|m|}(\cos\varphi)\sin(|m|\theta) & m < 0. \end{cases} \tag{A1}$$





Here $\bar{P}_{lm}$ are the associated Legendre polynomials (ALPs) of degree $l$ and order $m$, and the bar notation on the $\bar{P}_{lm}$ indicates
that they are '$4\pi$ normalised' (other normalisations are possible). Let the representation in spectral space be $\chi_{lm}$, and in real space be $x_{ij}$ (longitude $\theta_i$, co-latitude $\varphi_j$), which are related via the following linear combination:

$$x_{ij} = \sum_{l=0}^{L} \sum_{m=-l}^{l} \chi_{lm} Y_{lm}(\theta_i, \varphi_j), \tag{A2}$$

where $L$ is the chosen maximum degree. Equation (A2) is the mathematical form of the spectral transform. From the perspective of the transforms, the longitudes are best represented on a regularly spaced grid of $2L+1$ points, and the latitudes on a
Gaussian grid of $L+1$ points Errera and Ménard (2012); Wieczorek et al. (2018). This is the grid structure assumed here and the difference between this grid and the INVICAT grid is the reason for the $\mathbf{R}_{\mathrm{h}}$ operator in $\mathbf{B}_{\mathrm{sp}}^{1/2}$ in Eq. (7). The Gaussian grid enables exact quadrature (known as Gauss-Legendre quadrature, see Appendix A2).

In order to translate (A2) onto a computer, specifically using standard libraries, we do some rewriting. Substituting (A1) into (A2), separating the $l$ summation into $l=0$ and $l>0$ parts, and further separating the $m$ summation into $-l \le m \le -1$,
$m=0$, and $1 \le m \le l$ parts:

$$
\begin{aligned}
x_{ij} & = \chi_{00}\bar{P}_{00}(\cos\varphi_j) + \\
& \sum_{l=1}^{L}\left[\sum_{m=-l}^{-1}\chi_{lm}\bar{P}_{l|m|}(\cos\varphi_j)\sin(|m|\theta_i) + \sum_{m=0}^{l}\chi_{lm}\bar{P}_{lm}(\cos\varphi_j)\cos(m\theta_i)\right] \\
& = \chi_{00}\bar{P}_{00}(\cos\varphi_j) + \\
& \sum_{l=1}^{L}\left[\sum_{m=1}^{l}\chi_{l(-m)}\bar{P}_{lm}(\cos\varphi_j)\sin(m\theta_i) + \chi_{l0}\bar{P}_{l0}(\cos\varphi_j) + \sum_{m=1}^{l}\chi_{lm}\bar{P}_{lm}(\cos\varphi_j)\cos(m\theta_i)\right].
\end{aligned}
$$

The first and third terms can be combined, as can the second and fourth terms:

$$x_{ij} = \sum_{l=0}^{L}\chi_{l0}\bar{P}_{l0}(\cos\varphi_j) + \sum_{l=1}^{L}\sum_{m=1}^{l}\bar{P}_{lm}(\cos\varphi_j)\left[\chi_{l(-m)}\sin(m\theta_i) + \chi_{lm}\cos(m\theta_i)\right].$$

Further, let $\chi_{lm} = \chi_{lm}^{\mathrm{R}}$ ($0 \le m \le l$), and $\chi_{l(-m)} = \chi_{lm}^{\mathrm{I}}$ ($1 \le m \le l$):

$$x_{ij} = \sum_{l=0}^{L}\chi_{l0}^{\mathrm{R}}\bar{P}_{l0}(\cos\varphi_j) + \sum_{l=1}^{L}\sum_{m=1}^{l}\bar{P}_{lm}(\cos\varphi_j)\left[\chi_{lm}^{\mathrm{R}}\cos(m\theta_i) + \chi_{lm}^{\mathrm{I}}\sin(m\theta_i)\right]. \tag{A3}$$

This change of labelling means that we do not have to worry about negative wavenumber indices.
The double summation in Eq. (A3) first loops over $l$, and then loops over $m$ where the upper $m$ limit depends on $l$. This is summation order 1 in Fig. A1(a). In order to compute the above transform with standard software libraries, we first change the summations to the equivalent order 2 in Fig. A1(b):

$$x_{ij} = \sum_{l=0}^{L}\chi_{l0}^{\mathrm{R}}\bar{P}_{l0}(\cos\varphi_j) + \sum_{m=1}^{L}\sum_{l=m}^{L}\bar{P}_{lm}(\cos\varphi_j)\left[\chi_{lm}^{\mathrm{R}}\cos(m\theta_i) + \chi_{lm}^{\mathrm{I}}\sin(m\theta_i)\right]. \tag{A4}$$



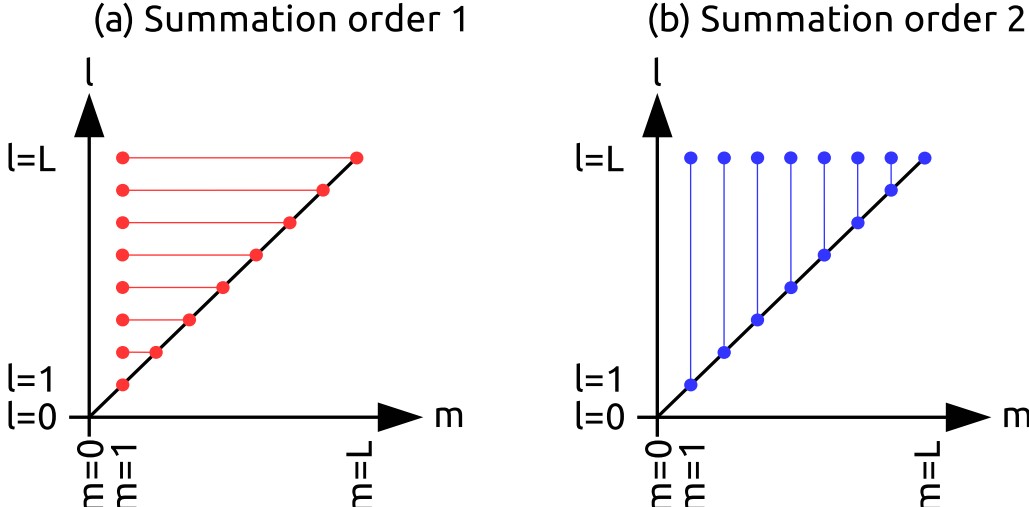

**Figure A1.** Orders of summations in $m$ and $l$ space as used in transforms (A3) and (A4). In order 1 (panel a), the sum over $l$ is done first, and in order 2 (panel b), the sum over $m$ is done first. Each order of summation is equivalent to the other as the same points are involved in the double summation.

This can then be split into a Legendre part and a Fourier part. First define the following:

$$\tilde{\chi}_m^R(\cos\varphi_j) \;=\; \sum_{l=m}^{L} \chi_{lm}^R \bar{P}_{lm}(\cos\varphi_j) \quad 0 \le m \le L \tag{A5a}$$

$$\tilde{\chi}_m^I(\cos\varphi_j) \;=\; \sum_{l=m}^{L} \chi_{lm}^I \bar{P}_{lm}(\cos\varphi_j) \quad 1 \le m \le L, \tag{A5b}$$

which leaves the rest of the transform as

$$x_{ij} = \tilde{\chi}_0^R(\cos\varphi_j) + \sum_{m=1}^{L} \left[ \tilde{\chi}_m^R(\cos\varphi_j) \cos(m\theta_i) + \tilde{\chi}_m^I(\cos\varphi_j) \sin(m\theta_i) \right]. \tag{A6}$$

The inputs to this spectral transform are the sets of coefficients in $m, l$-space $\chi_{lm}^R$ ($0 \le m \le L$, $m \le l \le L$), and $\chi_{lm}^I$ ($1 \le m \le L$, $m \le l \le L$). These are transformed to the intermediate sets of coefficients in $m, j$ space, $\tilde{\chi}_m^R(\cos\varphi_j)$ and $\tilde{\chi}_m^I(\cos\varphi_j)$ using Eqs. (A5) (the 'tilde' variables are half in spectral space, half in real space). In the code, these summations are performed explicitly using tables of ALPs pre-calculated from *SHTools* Wieczorek et al. (2018). The spectral transform is completed to $i, j$ space (complete real space) using the standard Fourier transform (A6), which is performed exactly by the routine *rfft1b* in the *fftpack* library Swarztrauber et al. (2016). Note that in data assimilation work, the transform from spectral to real space is part of the *forward transform* (since it is part of the string of operators that go from control space to observation space, $\mathbf{HB}^{1/2}$ in Eq. (4)), although readers may be more familiar with the transforms (A5) and (A6) being referred to an *inverse transforms*. This is an unfortunate clash of terminology.



## A2 The inverse spectral transform

The inverse spectral transform, $\mathbf{S}_h^{-1}$, changes the representation of a field increment from real space (a function of longitude, $\lambda_i$, and co-latitude, $\varphi_j$) to spectral space (a function of the total wavenumber, $l$, and zonal wavenumber, $m$, integers). The inverse of (A5) and (A6) is done in reverse order. Firstly, the $\tilde{\chi}_m^R$ and $\tilde{\chi}_m^I$ coefficients are found from standard Fourier transform formulae, which exploit the orthogonality of the sine and cosine functions:

$$\tilde{\chi}_0^R(\cos\varphi_j) = \frac{1}{2L+1}\sum_{i=0}^{2L}x_{ij} \tag{A7a}$$

$$\tilde{\chi}_m^R(\cos\varphi_j) = \frac{2}{2L+1}\sum_{i=0}^{2L}x_{ij}\cos m\theta_i \quad 1\leq m\leq L \tag{A7b}$$

$$\tilde{\chi}_m^I(\cos\varphi_j) = \frac{2}{2L+1}\sum_{i=0}^{2L}x_{ij}\sin m\theta_i \quad 1\leq m\leq L. \tag{A7c}$$

This is the inverse of (A6) and is performed by the routine *rfft1f* in the *fftpack* library. Next the $\chi_{lm}^R$ and $\chi_{lm}^I$ (fully spectral) coefficients are found from the formulae (derived below):

$$\chi_{l0}^R = \frac{1}{2}\sum_{j=0}^{L}\tilde{\chi}_0^R(\cos\varphi_j)\bar{P}_{l0}(\cos\varphi_j)\text{GaussWt}(\varphi_j) \quad 0\leq l\leq L \tag{A8a}$$

$$\chi_{lm}^R = \frac{1}{4}\sum_{j=0}^{L}\tilde{\chi}_m^R(\cos\varphi_j)\bar{P}_{lm}(\cos\varphi_j)\text{GaussWt}(\varphi_j) \quad 1\leq m\leq L \quad m\leq l\leq L \tag{A8b}$$

$$\chi_{lm}^I = \frac{1}{4}\sum_{j=0}^{L}\tilde{\chi}_m^I(\cos\varphi_j)\bar{P}_{lm}(\cos\varphi_j)\text{GaussWt}(\varphi_j) \quad 1\leq m\leq L \quad m\leq l\leq L, \tag{A8c}$$

where $\text{GaussWt}(\varphi_j)$ are the Gaussian weights (GWs), which are found (together with the ALPs themselves) from the *SHtools* package. The make up the inverse of (A5). The GWs allow co-latitude integrals of the following type to be discretised when values $f(\varphi_j)$ are on Gaussian latitude points $\varphi_j$ as shown:

$$\int_{\varphi=0}^{\pi}f(\varphi)\sin\varphi d\varphi = \sum_{j=0}^{L}f(\varphi_j)\text{GaussWt}(\varphi_j). \tag{A9}$$

Equations (A8) are derived from (A5) via the following orthogonality property of ALPs:

$$\int_{\varphi=0}^{\pi}\bar{P}_{lm}(\cos\varphi)\bar{P}_{l'm}(\cos\varphi)\sin\varphi d\varphi = 2(2-\delta_{0m})\delta_{ll'}, \tag{A10}$$

which, using (A9), becomes

$$\sum_{j=0}^{L}\bar{P}_{lm}(\cos\varphi_j)\bar{P}_{l'm}(\cos\varphi_j)\text{GaussWt}(\varphi_j) = 2(2-\delta_{0m})\delta_{ll'}. \tag{A11}$$





Multiplying (A5a) by $\bar{P}_{l'm}(\cos\varphi_j)\mathrm{GaussWt}(\varphi_j)$, summing over $j$, and then using (A11) gives

$$
\begin{aligned}
\sum_{j=0}^{L}\tilde{\chi}_m^{\mathrm{R}}(\cos\varphi_j)\bar{P}_{l'm}(\cos\varphi_j)\mathrm{GaussWt}(\varphi_j) &= \sum_{l=m}^{L}\chi_{lm}^{\mathrm{R}}\sum_{j=0}^{L}\bar{P}_{lm}(\cos\varphi_j)\bar{P}_{l'm}(\cos\varphi_j)\mathrm{GaussWt}(\varphi_j) \quad 0 \le m \le L \\
&= 2(2-\delta_{0m})\sum_{l=m}^{L}\chi_{lm}^{\mathrm{R}}\delta_{ll'} \\
&= 2(2-\delta_{0m})\chi_{l'm}^{\mathrm{R}} \qquad \text{for } m \le l' \le L.
\end{aligned}
$$

When $m=0$ this gives (A8a) and when $1 \le m \le L$ this gives (A8b). Now multiplying (A5b) by $\bar{P}_{l'm}(\cos\varphi_j)\mathrm{GaussWt}(\varphi_j)$, summing over $j$, and then using (A11) similarly gives

$$
\begin{aligned}
\sum_{j=0}^{L}\tilde{\chi}_m^{\mathrm{I}}(\cos\varphi_j)\bar{P}_{l'm}(\cos\varphi_j)\mathrm{GaussWt}(\varphi_j) &= \sum_{l=m}^{L}\chi_{lm}^{\mathrm{I}}\sum_{j=0}^{L}\bar{P}_{lm}(\cos\varphi_j)\bar{P}_{l'm}(\cos\varphi_j)\mathrm{GaussWt}(\varphi_j) \quad 1 \le m \le L \\
&= 2(2-\delta_{0m})\sum_{l=m}^{L}\chi_{lm}^{\mathrm{I}}\delta_{ll'} \\
&= 2(2-\delta_{0m})\chi_{l'm}^{\mathrm{I}} \qquad \text{for } m \le l' \le L.
\end{aligned}
$$

The $m=0$ case is of no interest, which leaves only the cases when $1 \le m \le L$, which gives (A8c).

**A3 The adjoint spectral transform**

The adjoint of the spectral transform (which is part of $\mathbf{B}^{\mathsf{T}/2}$) is needed to compute implied covariances ($\mathbf{B}^{1/2}\mathbf{B}^{\mathsf{T}/2}$) and the gradient of the cost function (5). Given that the adjoint operator is mathematically equivalent to the operator that propagates gradients (in the reverse direction to the forward counterparts), we use this equivalence as a means of deriving the adjoint versions of (A6) and (A5). Let variables with a hat represent gradients (of some unspecified function) with respect to that variable, namely $\hat{z}=\partial/\partial z$.

We will start with (A6), and look at the gradient with respect to each input component ($\tilde{\chi}_0^{\mathrm{R}}(\cos\varphi_j)$, $\tilde{\chi}_m^{\mathrm{R}}(\cos\varphi_j)$, and $\tilde{\chi}_m^{\mathrm{I}}(\cos\varphi_j)$) and exploit the chain rule in each case. The first input component of the forward transform (A6) is $\tilde{\chi}_0^{\mathrm{R}}(\cos\varphi_j)$. The gradient with respect to this variable, $\hat{\tilde{\chi}}_0^{\mathrm{R}}(\cos\varphi_j)$, is the output of the adjoint operation:

$$
\hat{\tilde{\chi}}_0^{\mathrm{R}}(\cos\varphi_j) = \sum_{i=0}^{2L}\frac{\partial x_{ij}}{\partial\tilde{\chi}_0^{\mathrm{R}}(\cos\varphi_j)}\hat{x}_{ij} = \sum_{i=0}^{2L}\hat{x}_{ij}, \tag{A12}
$$

where the partial derivative is found from (A6). The next group of input components to (A6) is $\tilde{\chi}_m^{\mathrm{R}}(\cos\varphi_j)$ ($1 \le m \le L$). The
gradient with respect to this variable, $\hat{\tilde{\chi}}_m^{\mathrm{R}}(\cos\varphi_j)$, is another output of the adjoint operation and is calculated are a similar way:

$$
\hat{\tilde{\chi}}_m^{\mathrm{R}}(\cos\varphi_j)\sum_{i=0}^{2L}\frac{\partial x_{ij}}{\partial\tilde{\chi}_m^{\mathrm{R}}(\cos\varphi_j)}\hat{x}_{ij} = \sum_{i=0}^{2L}\cos(m\theta_i)\hat{x}_{ij} \quad m \le l \le L. \tag{A13}
$$



The final group of input components is $\tilde{\chi}_m^{\mathrm{I}}(\cos\varphi_j)$ $(1 \leq m \leq L)$ and are calculated in a similar way:

$$\hat{\tilde{\chi}}_m^{\mathrm{I}}(\cos\varphi_j)\sum_{i=0}^{2L}\frac{\partial x_{ij}}{\partial \tilde{\chi}_m^{\mathrm{I}}(\cos\varphi_j)}\hat{x}_{ij} = \sum_{i=0}^{2L}\sin(m\theta_i)\hat{x}_{ij} \quad m \leq l \leq L. \tag{A14}$$

Notice that the adjoint operators (A12), (A13), and (A14) are similar to the inverse Fourier formulae (A7a), (A7b), and (A7c) respectively, but with no prefactors. This means that the *rfft1f* routine in the *fftpack* library can be used to perform the adjoint.

The next adjoint step is of (A5), and we will look at the gradient with respect to each input component ($\chi_{lm}^{\mathrm{R}}$ and $\chi_{lm}^{\mathrm{I}}$). The first group of input components of the forward transform (A5) is $\chi_{lm}^{\mathrm{R}}$:

$$\hat{\chi}_{lm}^{\mathrm{R}} = \sum_{j=0}^{L}\frac{\partial \tilde{\chi}_m^{\mathrm{R}}(\cos\varphi_j)}{\partial \chi_{lm}^{\mathrm{R}}}\hat{\tilde{\chi}}_m^{\mathrm{R}}(\cos\varphi_j) = \sum_{j=0}^{L}\bar{P}_{lm}(\cos\varphi_j)\hat{\tilde{\chi}}_m^{\mathrm{R}}(\cos\varphi_j) \quad 0 \leq m \leq L, \quad m \leq l \leq L. \tag{A15}$$

The final group of input components is $\chi_{lm}^{\mathrm{I}}$:

$$\hat{\chi}_{lm}^{\mathrm{I}} = \sum_{j=0}^{L}\frac{\partial \tilde{\chi}_m^{\mathrm{I}}(\cos\varphi_j)}{\partial \chi_{lm}^{\mathrm{I}}}\hat{\tilde{\chi}}_m^{\mathrm{I}}(\cos\varphi_j) = \sum_{j=0}^{L}\bar{P}_{lm}(\cos\varphi_j)\hat{\tilde{\chi}}_m^{\mathrm{I}}(\cos\varphi_j) \quad 1 \leq m \leq L, \quad m \leq l \leq L. \tag{A16}$$

Notice that the adjoint operators (A15) and (A16) are similar to the inverse Legendre formulae (A8b) and (A8c) respectively, but with different scaling (specifically no prefactors and no Gaussian weights).

The above steps are needed to perform the adjoint of the spectral transform in terms of separate Fourier and Legendre steps. It is also possible to write the adjoint of the whole horizontal transform (A2) in one go:

$$\hat{\chi}_{lm} = \sum_{j=0}^{L}\sum_{i=0}^{2L}Y_{lm}(\theta_i,\varphi_j)\hat{x}_{ij} \quad 0 \leq l \leq L, \quad -l \leq m \leq l, \tag{A17}$$

which is useful for the calibration of the horizontal transform (Appendix C1).

## Appendix B: Notes on the practical application of the operators in Eq. (7)

### B1 Application of the horizontal transform

The initial condition (ic) part of the control vector for use with the upper left part of Eq. (7) is a field that is a function of $l$, $m$, and height $z$, $\boldsymbol{\chi}_{c^0}(l,m,z)$. The $\mathbf{R}_{\mathrm{h}}\mathbf{S}_{\mathrm{h}}\boldsymbol{\Lambda}_{\mathrm{hc}}^{1/2}$ part of the CVT does the following. Let the $l$th diagonal element of $\boldsymbol{\Lambda}_{\mathrm{hc}}^{1/2}$ be $\boldsymbol{\Lambda}_{\mathrm{hc}}^{1/2}(l)$. The action of $\boldsymbol{\Lambda}_{\mathrm{hc}}^{1/2}$ is to multiply $\boldsymbol{\chi}_{c^0}(l,m,z)$ by $\boldsymbol{\Lambda}_{\mathrm{hc}}^{1/2}(l)$. The resulting fields then pass through the $\mathbf{S}_{\mathrm{h}}$ operator (separately for each $z$) resulting in a function of $\lambda,\varphi,z$. The reconfiguration operator $\mathbf{R}_{\mathrm{h}}$ interpolates this horizontally from the $\lambda,\varphi$ grid imposed by *SHTools* to the $\lambda,\phi$ grid of INVICAT.

The flux part of the control vector for use with the lower right part of Eq. (7) is a field that is a function of $l$, $m$, and time $t$, $\boldsymbol{\chi}_{\rho}(l,m,t)$. The $\mathbf{R}_{\mathrm{h}}\mathbf{S}_{\mathrm{h}}\boldsymbol{\Lambda}_{\mathrm{h}\rho}^{1/2}$ part of the CVT does a similar thing as for the initial condition, apart from the result being a function of $\lambda,\phi,t$.





### B2  Application of the vertical transform

Following on from the horizontal and reconfiguration transforms for the ic fields, the vertical transform, $\boldsymbol{\Xi}^{-1}\mathbf{F}_{vc}\boldsymbol{\Lambda}_{vc}^{1/2}\mathbf{F}_{vc}^{\mathsf{T}}$, acts

on fields that are a function of $\lambda, \phi, z$. $\boldsymbol{\Xi}$, $\mathbf{F}_{vc}$ and $\boldsymbol{\Lambda}_{vc}$ are each $n_z \times n_z$ matrices ($\boldsymbol{\Xi}$ and $\boldsymbol{\Lambda}_{vc}$ are diagonal). $\mathbf{F}_{vc}^{\mathsf{T}}$ projects onto vertical modes (the eigenvectors), $\boldsymbol{\Lambda}_{vc}^{1/2}$ scales the projections, and $\mathbf{F}_{vc}$ projects from the vertical modes back to $z$. This is done for each horizontal position separately. There is a different $\boldsymbol{\Lambda}_{vc}^{1/2}$ and $\boldsymbol{\Xi}$ for each latitude. The output of the vertical transform is also a function of $\lambda, \phi, z$.

### B3  Application of the temporal transform

Following on from the horizontal and reconfiguration transforms for the flux fields, the temporal transform, $\mathbf{F}_{t\rho}\boldsymbol{\Lambda}_{t\rho}^{1/2}\mathbf{F}_{t\rho}^{\mathsf{T}}$, acts on fields that are a function of $\lambda, \phi, t$. $\mathbf{F}_{t\rho}$ and $\boldsymbol{\Lambda}_{t\rho}$ are each $(T+1) \times (T+1)$ matrices, and $\boldsymbol{\Lambda}_{t\rho}$ is diagonal. $\mathbf{F}_{t\rho}^{\mathsf{T}}$ projects onto temporal modes (the eigenvectors), $\boldsymbol{\Lambda}_{t\rho}^{1/2}$ scales the projections, and $\mathbf{F}_{t\rho}$ projects from the temporal modes back to $t$. This is done for each horizontal position separately. The $\boldsymbol{\Lambda}_{t\rho}^{1/2}$ is the same for each horizontal position. The output of the temporal transform is also a function of $\lambda, \phi, t$.

### Appendix C:  Determining the transform matrices


Equation (7) is the form of the spectral-based control variable transform. The objects to be determined are the following: $\boldsymbol{\Xi}$, $\mathbf{F}_{vc}$, $\boldsymbol{\Lambda}_{vc}$, $\boldsymbol{\Lambda}_{hc}$, $\mathbf{F}_{t\rho}$, $\boldsymbol{\Lambda}_{t\rho}$, and $\boldsymbol{\Lambda}_{h\rho}$ (other objects in that equation are either standard transforms – such as the spherical transform, or take assumed values – see Sect. 3.2). The calibration procedures for determining the above objects are described here.

### C1  Determining the 'variance spectra' of the horizontal transforms


The horizontal transforms for the ic and the flux fields are each described in the form $\mathbf{S}_h\boldsymbol{\Lambda}_h^{1/2}$, where $\mathbf{S}_h$ is the spectral transform (spherical harmonics to latitude/longitude), $\boldsymbol{\Lambda}_h^{1/2} = \boldsymbol{\Lambda}_{hc}^{1/2}$ for the ic, and $\boldsymbol{\Lambda}_h^{1/2} = \boldsymbol{\Lambda}_{h\rho}^{1/2}$ for the flux. This section describes how $\boldsymbol{\Lambda}_h$ can be determined given a prescribed form of the horizontal correlations.

The implied correlation is found by $\mathbf{S}_h\boldsymbol{\Lambda}_h^{1/2}\left(\mathbf{S}_h\boldsymbol{\Lambda}_h^{1/2}\right)^{\mathsf{T}} = \mathbf{S}_h\boldsymbol{\Lambda}_h\mathbf{S}_h^{\mathsf{T}}$. Using the expanded form of the spectral transform

(A2) and the adjoint (A17), the result of acting with $\mathbf{S}_h\boldsymbol{\Lambda}_h\mathbf{S}_h^{\mathsf{T}}$ on a field that is a function of longitude $\lambda_i$ and co-latitude $\varphi_j$ ($f(\lambda_i, \varphi_j)$), giving $f'(\lambda_{i'}\varphi_{j'})$ is

$$f'(\lambda_{i'}\varphi_{j'}) = \underbrace{\sum_{l=0}^{L}\sum_{m=-l}^{l} Y_{lm}(\theta_{i'}, \varphi_{j'})}_{\mathbf{S}_h}\underbrace{\boldsymbol{\Lambda}_h(l,m)}_{\boldsymbol{\Lambda}_h}\underbrace{\sum_{j=0}^{L}\sum_{i=0}^{2L} Y_{lm}(\theta_i, \varphi_j)}_{\mathbf{S}_h^{\mathsf{T}}} f(\lambda_i, \varphi_j).$$

For simplicity, we allow $\boldsymbol{\Lambda}_h$ to be a function of total wavenumber, $l$, only:

$$f'(\lambda_{i'}\varphi_{j'}) = \sum_{j=0}^{L}\sum_{i=0}^{2L}\sum_{l=0}^{L}\boldsymbol{\Lambda}_h(l)\sum_{m=-l}^{l} Y_{lm}(\theta_{i'}, \varphi_{j'})Y_{lm}(\theta_i, \varphi_j)f(\lambda_i, \varphi_j). \tag{C1}$$



The addition theorem of spherical harmonics (see e.g. Appendix A3, Eq. (A9) of Errera and Ménard (2012)) is

$$\sum_{m=-l}^{l} Y_{lm}(\theta_{i'}, \varphi_{j'}) Y_{lm}(\theta_i, \varphi_j) = \sqrt{2l+1} \bar{P}_{l0}(\cos\alpha), \tag{C2}$$

where $\alpha$ is the great circle angular separation between positions $(\theta_{i'}, \varphi_{j'})$ and $(\theta_i, \varphi_j)$. Substituting (C2) into (C1) gives

$$f'(\lambda_{i'}\varphi_{j'}) = \sum_{j=0}^{L}\sum_{i=0}^{2L}\underbrace{\sum_{l=0}^{L}\mathbf{\Lambda}_{\mathrm{h}}(l)\sqrt{2l+1}\bar{P}_{l0}(\cos\alpha)}_{\text{implied correlation matrix element}} f(\lambda_i, \varphi_j),$$

and the implied correlation between $(\theta_{i'}, \varphi_{j'})$ and $(\theta_i, \varphi_j)$ due to this horizontal transformation is the part indicated above:

$$C_{\mathrm{h}}(\lambda_{i'}\varphi_{j'}; \lambda_i, \varphi_j) = C_{\mathrm{h}\alpha}(\alpha) = \sum_{l=0}^{L} \mathbf{\Lambda}_{\mathrm{h}}(l)\sqrt{2l+1}\bar{P}_{l0}(\cos\alpha). \tag{C3}$$

Notice that the right hand side is a function only of the separation between the two positions, $\alpha$, and not on their individual values nor their relative orientation. These properties are called homogeneity and isotropy respectively. Let $C_{\mathrm{h}\alpha}(\alpha)$ be a prescribed function, e.g. the SOAR function $\{1+|d/\xi|\}\exp(-d/\xi)$, where $d = \alpha\pi R_{\mathrm{E}}/180$, $\alpha$ is in degrees, $R_{\mathrm{E}}$ is the radius of the Earth, and $\xi$ is the chosen length-scale of the correlation (specified in the same units as $R_{\mathrm{E}}$). Multiply each side of (C3)

by $\bar{P}_{l'0}(\cos\alpha)\sin\alpha$, and integrate over $\alpha = 0, \pi$:

$$\int_{\alpha=0}^{\pi} C_{\mathrm{h}\alpha}(\alpha)\bar{P}_{l'0}(\cos\alpha)\sin\alpha = \sum_{l=0}^{L} \mathbf{\Lambda}_{\mathrm{h}}(l)\sqrt{2l+1} \int_{\alpha=0}^{\pi} \bar{P}_{l0}(\cos\alpha)\bar{P}_{l'0}(\cos\alpha)\sin\alpha.$$

The orthogonality property (A10) is then used:

$$\int_{\alpha=0}^{\pi} C_{\mathrm{h}\alpha}(\alpha)\bar{P}_{l'0}(\cos\alpha)\sin\alpha = \sum_{l=0}^{L} \mathbf{\Lambda}_{\mathrm{h}}(l)\sqrt{2l+1}\, 2\delta_{ll'} = 2\sqrt{2l'+1}\mathbf{\Lambda}_{\mathrm{h}}(l').$$

The horizontal variance spectrum is then given below in continuous form and then in discrete form using (A9):

$$\mathbf{\Lambda}_{\mathrm{h}}(l) = \frac{1}{2\sqrt{2l+1}} \int_{\alpha=0}^{\pi} C_{\mathrm{h}\alpha}(\alpha)\bar{P}_{l0}(\cos\alpha)\sin\alpha$$

$$= \frac{1}{2\sqrt{2l+1}} \sum_{j=0}^{L} C_{\mathrm{h}\alpha}(\alpha_j)\bar{P}_{l0}(\cos\alpha_j)\mathrm{GaussWt}(\alpha_j). \tag{C4}$$

It is important to ensure that $C_{\mathrm{h}\alpha}(0) = 1$ (which may not be true due to numerical errors in the above procedure). Once $\mathbf{\Lambda}_{\mathrm{h}}(l)$ is computed with (C4), $C_{\mathrm{h}\alpha}(0) = 1$ is ensured by multiplying $\mathbf{\Lambda}_{\mathrm{h}}(l)$ by $1/\left(\sum_{l=0}^{L} \mathbf{\Lambda}_{\mathrm{h}}(l)\sqrt{2l+1}\bar{P}_{l0}(1)\right)$.

### C2    Determining the vertical transform for the initial concentration

The vertical transform is described by the $n_z \times n_z$ matrices $\mathbf{\Xi}$, $\mathbf{F}_{\mathrm{vc}}$ and $\mathbf{\Lambda}_{\mathrm{vc}}$ (see bullet points 3 and 4 of Sect. 3.2), where $n_z$ is the number of vertical levels. In principle, there can be a different set of such vertical matrices for each horizontal position.





For simplicity, and because we have only limited knowledge of what the vertical correlations should be, we use just one set of eigenvectors (in $\mathbf{F}_{vc}$), found from a global vertical correlation matrix (see below), but there are multiple sets of 'eigenvalue' matrices (now labelled as $\mathbf{\Lambda}_{vc}(\phi)$) – one for each latitude position $\phi$.

Suppose that we have target vertical correlation matrices, found from data at each latitude $\phi$, $\mathbf{C}^{v}(\phi)$. Further suppose that we also have a vertical correlation matrix found from globally averaged data, $\mathbf{C}^{vg}$. Let $\mathbf{F}_{vc}$ be the eigenvectors of $\mathbf{C}^{vg}$. The matrices $\mathbf{\Lambda}_{vc}(\phi)$ are found from the diagonal elements of the projection of $\mathbf{C}^{v}(\phi)$ into the space of $\mathbf{F}_{vc}$, namely $\mathbf{\Lambda}_{vc}(\phi) = \mathrm{diag}\left(\mathbf{F}_{vc}^{\mathsf{T}}\mathbf{C}^{v}(\phi)\mathbf{F}_{vc}\right)$, where $\mathrm{diag}(\bullet)$ sets off-diagonal elements to zero. Note that, although the $\mathbf{\Lambda}_{vc}(\phi)$ are referred to as 'eigenvalue' matrices, they are only approximations to the eigenvalues at latitude $\phi$.

The implied correlation matrix that this transform is intended to represent, $\left(\mathbf{F}_{vc}\mathbf{\Lambda}_{vc}^{1/2}(\phi)\mathbf{F}_{vc}^{\mathsf{T}}\right)\left(\mathbf{F}_{vc}\mathbf{\Lambda}_{vc}^{1/2}(\phi)\mathbf{F}_{vc}^{\mathsf{T}}\right)^{\mathsf{T}}$, is not guaranteed to be a strict correlation matrix due to the fact that $\mathbf{\Lambda}_{vc}(\phi)$ are not the exact eigenvalues (the explicit $\phi$ is now dropped for brevity). This is compensated for with the ($\phi$-dependent and diagonal) $\mathbf{\Xi}^{-1}$ operator in (7) is chosen such that $\left(\mathbf{\Xi}^{-1}\mathbf{F}_{vc}\mathbf{\Lambda}_{vc}^{1/2}\mathbf{F}_{vc}^{\mathsf{T}}\right)\left(\mathbf{\Xi}^{-1}\mathbf{F}_{vc}\mathbf{\Lambda}_{vc}^{1/2}\mathbf{F}_{vc}^{\mathsf{T}}\right)^{\mathsf{T}}$ has unit diagonal elements. This is achieved by setting the $i$th diagonal element of $\mathbf{\Xi}$ to $\mathbf{\Xi}_{ii} = \sqrt{\sum_j \left[\mathbf{F}_{vc}\right]_{ij}^2 \left[\mathbf{\Lambda}_{vc}\right]_{jj}}$.

The correlation matrices are formed from 10 years (1995-2004) of methane forecasts, first detrended to correct for methane trends (even though there was little methane growth over this period, Rigby et al. (2008)).

### C3  Determining the temporal transform for the flux

The temporal transform is described by the $(T+1) \times (T+1)$ matrices $\mathbf{F}_{t\rho}$ and $\mathbf{\Lambda}_{t\rho}$ (see bullet point 8 of Sect. 3.2). First a $(T+1) \times (T+1)$ correlation matrix $\mathbf{C}^{t}$ is formed with matrix elements $\mathbf{C}_{ij}^{t} = \{1 + |t_i - t_j|/\tau\}\exp\left(-|t_i - t_j|/\tau\right)$, where $t_i$
and $t_j$ are times (in months), and $\tau$ is the timescale. $\mathbf{F}_{t\rho}$ and $\mathbf{\Lambda}_{t\rho}$ are, respectively, eigenvectors and eigenvalues of $\mathbf{C}^{t}$.

### Appendix D: Comparison of costs of an explicit vs a spectral scheme

This appendix illustrates how the cost of the spectral $\mathbf{B}$-matrix scheme scales with the number of grid points, and how that cost compares to the brute-force option of constructing a full the $\mathbf{B}$-matrix explicitly and finding its square-root using eigen-decomposition. The latter is used with some systems, e.g. Chevallier et al. (2007). The costs are estimated as a function of
the maximum total wavenumber, $L$, which is associated with $2L+1$ longitudes and $L+1$ latitudes. Since we are interested in scaling only, these are each approximated to $L$, so the number of horizontal grid points scales as $\sim L^2$. Table D1 shows how various components of the $\mathbf{B}$-matrix model scale with $L$, the number of vertical levels, $n_z$, and the number of months, $T+1$. Costs are shown with respect to computing the object using eigen-decomposition (which has to be done only once, off-line column 3), and using the object within the minimisation (column 4). Row 1 refers to a diagonal representation of $\mathbf{B}$, row 2
refers to a full representation of $\mathbf{B}$, and rows 3 to 12 refer to the spectral representation (see Sect. 3.2). The costs are shown graphically in Fig. D1. It is clear to see how the explicit representation of $\mathbf{B}$ (blue) becomes unaffordable for large $L$, especially in setting-up the system by computing its eigenvalues and eigenvectors (continuous blue line), while the spectral representation remains affordable even for large $L$ (the current study has $L = 32$).



**Table D1.** How the cost of various $\mathbf{B}$-matrix model components scale with $L$, $n_z$, and $T$. The third column reflects the cost of computing the component, where that requires the eigenvalues and eigenvectors of a matrix of size $n$ (the computation is assumed to scale as $\mathcal{O}(n^3)$). The fourth column reflects the cost of storage and use of the component. In the case of $\mathbf{S}_h$, this has a part which is a fast Fourier transform, which scales as $(2L+1)\log_2(2L+1) \approx 2L\log_2(2L)$.

| | Object | How determination of related objects scale with system size | How cost of use and storage scale with system size |
|---|---|---|---|
| 1 | $\mathbf{B}_d$ | — | $L^2[n_z+(T+1)]$ |
| 2 | full $\mathbf{B}$-matrix | $\mathbf{B}^{1/2}$: $\left\{L^2[n_z+(T+1)]\right\}^3$ | $\left\{L^2[n_z+(T+1)]\right\}^2$ |
| 3 | $\mathbf{\Sigma}_c^b$ | — | $L^2 n_z$ |
| 4 | $\mathbf{\Sigma}_{\rho^0}^b,\ldots,\mathbf{\Sigma}_{\rho^T}^b$ | — | $L^2(T+1)$ |
| 5 | $\mathbf{\Lambda}_{hc}^{1/2}$ | $L^2 n_z$ | $Ln_z$ |
| 6 | $\mathbf{S}_h$ | — | $(n_z+T+1)L^3 \times [1+2\log_2(2L)]$ |
| 7 | $\mathbf{R}_h$ | — | negligible |
| 8 | $\mathbf{F}_{vc}$ | $\mathbf{F}_{vc}, \mathbf{\Lambda}_{vc}^{1/2}: Ln_z^3$ | $L^2 n_z^2$ |
| 9 | $\mathbf{\Lambda}_{vc}^{1/2}$ | | $Ln_z$ |
| 10 | $\mathbf{\Xi}^{-1}$ | — | negligible |
| 11 | $\mathbf{F}_{t\rho}$ | $(T+1)^3$ | $(T+1)^2 L^2$ |
| 12 | $\mathbf{\Lambda}_{t\rho}^{1/2}$ | | $(T+1)L^2$ |

*Author contributions.* RNB designed the covariance schemes, added the extra code to INVICAT, ran the tests, did the analysis, and lead the paper writing. CW advised on the INVICAT set-up, provided the existing INVICAT system and data, and wrote parts of the paper.

*Competing interests.* The authors declare that there are no competing interests in relation to this work.

*Acknowledgements.* The authors are supported by the National Centre for Earth Observation (NCEO, contract number PR140015). The experiments performed in the work were done on the Jasmin supercomputer operated by the UK Science and Technology Facilities Council, the Centre for Environmental Data Analysis, and the Scientific Computing Department. The authors would like to thank Wuhu Feng (University of Leeds) for assistance in getting INVICAT to run on Jasmin and Bill Collins (University of Reading) for providing advice on the manuscript. The authors thank the authors of the software packages LAPACK, SHTOOLS, FFTPACK, and M1QN3.



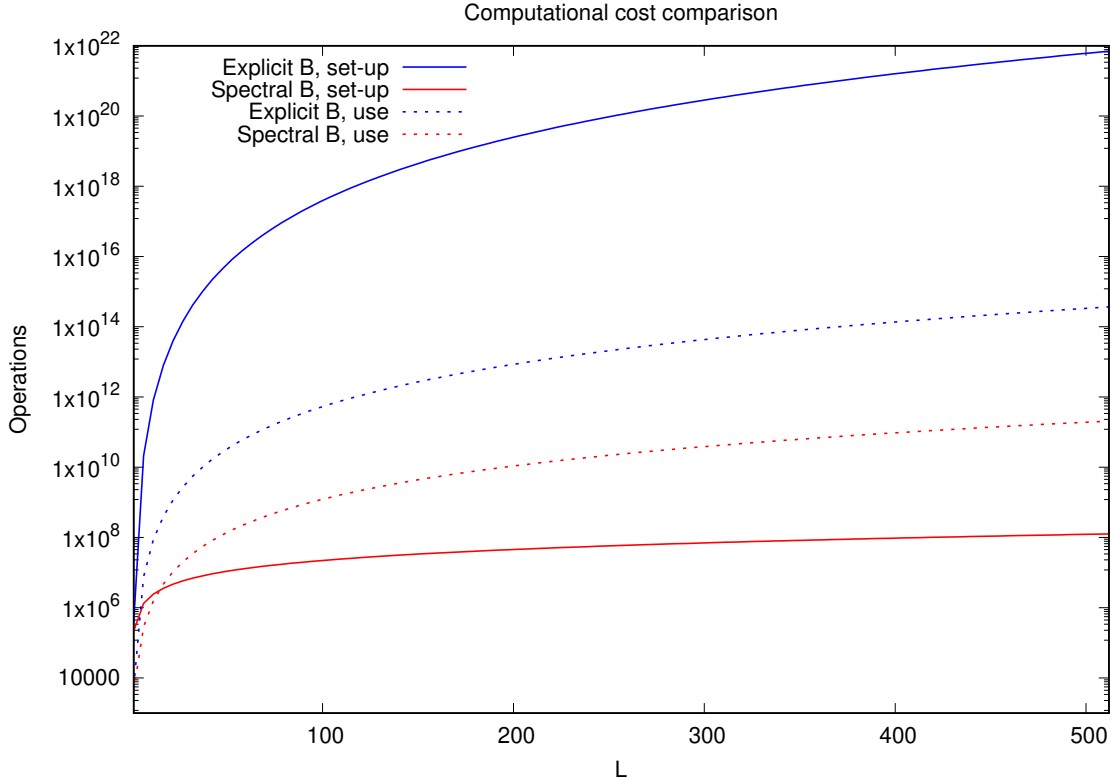

**Figure D1.** Plot of how the cost of various **B**-matrix model components scale with the total wavenumber, $L$. The continuous lines show how the cost of computing the components of **B** scale with $L$ (i.e. set-up costs, third column of Table D1), and the dashed lines show how the cost of using the components scale (fourth column). The blue lines are for the explicit **B**-matrix representation (row 2 in Table D1) and the red lines are for the spectral **B**-matrix representation (sum of rows 3 to 12). All curves assume $n_z = 60$ vertical levels and $T = 12$ months.

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
