# Peer review of "Inverse modelling for surface methane flux estimation with 4DVar: impact of a computationally efficient representation of a non-diagonal B-matrix in INVICAT v4"

_EGUsphere, 2024_

## Author Comment (AC1)

**Response to Anonymous Referee #1 (RC1)**

March 2024

We would like to thank Referee #1 for his/her comments on our work. Referee #1 has raised some interesting points, which we would like to deal with as early as possible, especially in case her/her comments influence other reviewers. We believe this response can resolve the apparent paradoxes that the Referee raises. We reproduce quotes from Referee #1's report (in italics), and respond.

- *Without being new, the article is interesting, stimulating, educational and generally well written.*

  - Thank you. We do believe, however, that the article is new as this is the first time (to our knowledge) that a spectral scheme has been implemented in a full flux inversion system.

- *The first paradox of the presentation is the motivation: "The spectral method is very efficient. It is applicable to systems with very high resolutions, where existing methods that explicitly represent the B-matrix would not be feasible (Appendix D)" (l. 377-379 and similar sentences in the rest of the text). Actually, looking at Figure D1 about the cost of the two approaches, one can see that the authors' illustration is on the lower end (L = 32, small resolution), while current results with an explicit representation reach the high end (L seems to be about 400 in doi:10.22541/au.171052488.85903583/v1). Where is the better efficiency argued in l. 66?*

  - The tests done in the paper are indeed for low resolution $L = 32$, but these are purely to show that the method works.

  - Looking at Fig. D1 for $L \sim 400$, the spectral method is 3-4 orders of magnitude more efficient to use than the explicit approach (compare dashed lines in Fig. D1), and even more so to setup (assuming an eigenvalue decomposition is used to find the square-root of the B-matrix – compare the continuous lines in Fig. D1).

- *The authors seem to ignore that the explicit representation is simplified by numerous zero correlations in the 2D flux errors: no space-time correlations, ..., no correlations between land and ocean for surface flux errors, ...*

  - Thank you. This was an oversight, and we believe the paradox can be resolved. We have revisited Appendix D concerning the costs of the spectral method compared to explicit representations of the B-matrix. By "explicit", we have now included three possibilities of explicit representation of the B-matrix (rather than one in the manuscript). (i) An explicit representation of B in the entire state space, which has $n = n_x n_y n_z + n_x n_y (T+1)$ elements ($n_x$ longitudes, $n_y$ latitudes, $T+1$ months). This B-matrix has $n^2$ elements; this is what is meant by the explicit matrix in the current manuscript. Our oversight was to neglect two more compact alternatives, which we believe, especially the latter, will address Referee #1's point. (ii) Separate explicit representations of the parts of the B-matrix that associated with the initial concentration, $c$, and with the flux, $\rho$. These B-matrices have $(n_x n_y n_z)^2$ and $(n_x n_y (T+1))^2$ elements respectively. (iii) The same as (ii), but where the vertical/horizontal and temporal/horizontal parts of the correlations are separable (e.g. total flux correlation = temporal correlation × horizontal correlation), which leads to use of the Kronecker product mentioned by the Referee. Although (ii) and (iii) are more efficient than (i), they are still much less efficient than the spectral method. We have shown the workings in a possible replacement to Table D1 and to Fig. D1 below.

  - For example, running an inversion at the resolution of ERA-5 reanalysis ($1440 \times 720$, corresponding to $L \approx 720$) let's compare the cost of using explicit form (iii) above (the most efficient of the explicit representations) to the cost of the spectral method. Let us put some numbers to our argument and consider the flux field only. The following are costs associated with setting-up the square-root matrices:
    * Explicit form (iii) for $\rho$: $\sim 10^{18}$ operations.
    * Spectral form for $\rho$: $\sim 10^7$ operations.

- The following are costs associated with each variational iteration:
  * Explicit form (iii) for $\rho$: $\sim 10^{13}$ operations.
  * Spectral form for $\rho$: $\sim 10^{7}$ operations.
- We hope these numbers provide a clear justification for the usefulness of the spectral method at high resolutions.

- *... (hidden cost of the remark made in l. 407-408 that suggests duplicating the control vector when using the spectral method)*

  - We think the above remark concerns how our spectral method can be adapted to decouple land and sea points, by duplicating the control vector. Although this duplicating doubles the cost of the spectral method, it is an almost negligible increase compared to any of the explicit schemes mentioned.

- *... the authors find that assigning spatial prior error correlations for the initial state degrades the inversion (see their embarrassed explanation in l. 417-420).*

  - We are actually intrigued, rather than embarrassed about this finding. This result will hopefully spark some more investigation concerning the role of biases between the observations and the forecasts, so we can make this clearer in any revision.

- *Atmospheric inversions suffer from edge effects and it is usual to cut off both ends: in contrast to NWP, obtaining the optimal initial state is not strategic and therefore the representation of its prior uncertainty can be simplified.*

  - We wanted to keep the edge effect in order to study the effect of changing the representation of the B-matrix for the initial concentration. Flux inversion is fundamentally affected by the initial conditions, so this part of the B-matrix deserves some attention. Whether the ends are cut or not in practice is a concern of any application of the method over and above the testing done in this paper.

- *The second paradox of the paper is related. The objective of the method is to facilitate the resolution increase, but the detail of the increments is blurred by the horizontal reconfiguration operator Rh.*

  - This blurring is only marginal, and anyway reduces with increased resolution (interpolation from the high-resolution grid required by the spectral transform to another high-resolution grid required by the model), so we do not regard this as a significant issue, or a paradox. In fact it would not be needed with a model that has the same grid as the spectral transform.

- There are other comments of Referee #1 that he/she describes as minor. These can be dealt with in any revision, if requested by the Editor.

| | Object | How determination of related objects scale with system size | How cost of use and storage scale with system size |
|---|---|---|---|
| 1 | $\mathbf{B}_d$ | — | $n_x n_y\,[n_z + T + 1]$ |
| 2 | explicit $\mathbf{B}$-matrix (i) ($c,\rho$ coupled) | $\mathbf{B}^{1/2}$: $\{n_x n_y\,[n_z + (T+1)]\}^3$ | $\{n_x n_y\,[n_z + (T+1)]\}^2$ |
| 3 | explicit $\mathbf{B}$-matrix (ii) ($c,\rho$ uncoupled) | $\{n_x n_y n_z\}^3 + \{n_x n_y (T+1)\}^3$ | $\{n_x n_y n_z\}^2 + \{n_x n_y (T+1)\}^2$ |
| 4 | explicit separable $\mathbf{B}$-matrix (iii) ($c,\rho$ uncoupled) | $(T+1)^3 + (n_x n_y)^3 + n_z^3 + (n_x n_y)^3$ | $n_x n_y (T+1) + (n_x n_y)^2(T+1) + (T+1)^2 n_x n_y + n_x n_y n_z + (n_x n_y)^2 n_z + n_z^2 n_x n_y$ |
| 5 | $\mathbf{\Sigma}_c^{\mathrm{b}}$ | — | $n_x n_y n_z$ |
| 6 | $\mathbf{\Sigma}_{\rho^0}^{\mathrm{b}}, \ldots, \mathbf{\Sigma}_{\rho^T}^{\mathrm{b}}$ | — | $n_x n_y (T+1)$ |
| 7 | $\mathbf{\Lambda}_{\mathrm{h}c}^{1/2}$ | $L^2 n_z$ | $L n_z$ |
| 8 | $\mathbf{\Lambda}_{\mathrm{h}\rho}^{1/2}$ | $L^2(T+1)$ | $L(T+1)$ |
| 9 | $\mathbf{S}_{\mathrm{h}}$ | — | $(T+1+n_z)\times [(L+1)L + (2L+1)\log_2(2L)]$ |
| 10 | $\mathbf{R}_{\mathrm{h}}$ | — | negligible |
| 11 | $\mathbf{F}_{vc}$ | $\mathbf{F}_{vc}, \mathbf{\Lambda}_{vc}^{1/2}$: $n_y n_z^3$ | $2 n_x n_y n_z^2$ |
| 12 | $\mathbf{\Lambda}_{vc}^{1/2}$ | | $n_x n_y n_z$ |
| 13 | $\mathbf{\Xi}^{-1}$ | — | negligible |
| 14 | $\mathbf{F}_{\mathrm{t}\rho}$ | $(T+1)^3$ | $2 n_x n_y (T+1)^2$ |
| 15 | $\mathbf{\Lambda}_{\mathrm{t}\rho}^{1/2}$ | | $n_x n_y (T+1)$ |

Tab. 1: Adapted from Table D1 in the paper. How the cost of various $\mathbf{B}$-matrix model components scale with $L$, $n_z$, and $T$. The 'explicit' schemes are given in terms of the number of longitudes and latitudes, which are related to $L$ via $n_x = 2L + 1$, $n_y = L + 1$. The third column reflects the cost of computing the component, where that requires the eigenvalues and eigenvectors of a matrix of size $n$ (the computation is assumed to scale as $\mathcal{O}(n^3)$). The fourth column reflects the cost of storage and use of the component in each variational iteration. In the case of $\mathbf{S}_{\mathrm{h}}$, this has a part which is a fast Fourier transform, which scales as $(2L+1)\log_2(2L+1)$.

[Figure]

Fig. 1: (new Fig. D1) Plot of how the cost of various **B**-matrix model components scale with the total wavenumber, $L$. The continuous lines show how the cost of computing the components of **B** scale with $L$ (i.e. setup costs, third column of Table 1), and the dashed lines show how the cost of using the components scale in each variational iteration (fourth column). The blue, purple, and green lines are respectively for the explicit **B**-matrix representations (i), (ii), and (iii) (rows 2, 3, and 4 in the Table), the red lines are for the spectral **B**-matrix representation (sum of rows 5 to 15), and the gold line is for the diagonal **B**-matrix (row 1 in the Table). All curves assume $n_z = 100$ vertical levels and $T = 12$ months.

---

## Author Comment (AC2)

**Response to Anonymous Referee #2 (RC2)**

July 2024

We would like to thank Referee #2 for his/her careful and balanced comments on our work and for their support for publication of this work. We reproduce quotes from Referee #2's report (in italics), and respond.

- Before the specific comments: *Whether it's actually practical is not clear, and the demo seems like an odd choice since it is so poorly observationally constrained.*

  - There are two ways that the practicality of a method can be defined. One is whether it is appropriate/applicable, and the other is whether it is efficient for use. We believe that the method set out in our paper is practical under both these definitions.
    On the first definition, the spectral method is associated with homogeneous correlations. While this property is not obeyed strictly in geophysical systems, it is a pragmatic assumption, which has traditionally been used successfully in many geophysical data assimilation problems. It is a first stab at the spectral method in a flux inversion problem. The paper does point out that the homogeneous property can be relaxed by using two spectral fields and masks, specifically to decouple land and sea points. Many other works in flux inversion assume either homogeneous and isotropic correlation functions (apart from avoiding land/sea coupling) for the flux, or assume that flux errors are uncorrelated, which are all approximations.
    On the second definition, the spectral method is extremely efficient (and would remain so using the two spectral fields to decouple land and sea). This is a very clear conclusion of the work.

  - The experiments shown, as the reviewer states, are just demos. The aim is to assimilate only in-situ data to avoid complications of satellite data. Although the available observations are relatively sparse, they are spread around the globe and we do believe are adequate to help validate the new method.

- Point 1: *Lines 38-40 ... It is not really fair to say that it is limited 'to a relatively small number of large-area surface regions' because the Jacobian matrix can be computed as an embarrassingly parallel problem ...*

  - Many thanks. We will include the reviewer's comments in a requested revision.

- Point 2: *Line 42 should also mention the LETKF approach used by Myazaki at JPL, ..., which has similarity to the transform done here.*

  - We will include this reference in a revision, but we don't believe that the LETKF data assimilation method uses the transform described in our paper.

- Point 3: *Lines 43-46 There should be some mention that the size of the state vector is limited not only by computational resources but also by information content. A problem with 4DVAR and EnKF methods is that information content is not directly characterized.*

  - Thank you for these interesting comments. We don't agree that the information content of the measurements should *necessarily* be a limiting factor in the size of the state vector, especially in the case of a non-diagonal B-matrix (see https://doi.org/10.5194/acp-16-14371-2016). Even if there are $n$ elements to the state vector, and only $m \ll n$ observations (or even fewer degrees of freedom of the signal), a large state vector may be necessary, for instance, to provide enough resolution for the model to behave realistically. Information content may be characterised by the reduction in the error covariance from the prior to the posterior. This information is difficult to get from 4DVar (although not impossible), but easier to get from EnKF methods. The degrees of freedom of the signal is another measure of the information content. This is simply twice the value of the background term evaluated at the analysis, so is available immediately from 4DVar.

- Point 4: *Line 60 another problem in empirical construction of B is ensuring that it is positive definite – this typically requires massaging the matrix after empirical construction of the off-diagonals.*

    – Indeed. A ridge regression (to ensure positive-definiteness) was necessary only for the error covariances of the prior methane initial conditions in the vertical direction. This is only a very minor part of this work anyway. The remaining covariance components are guaranteed to be positive-definite (see also point 6 below).

- Point 5: *Line 131 why does the change in grid-box size with latitude matter?*

    – For a diagonal B-matrix, the effective correlation length-scales are not zero, but of the order of the size of each grid box (and with structure functions that are steps). Hence as the grid box sizes of the TOMCAT model change with latitude, a diagonal B-matrix naturally imposes longer length-scales, which may not be realistic. This was only mentioned in passing in the manuscript, and can be explained in a revision.

- Point 6: *Line 152 will Bsp always be positive definite?*

    – Yes, we believe so, as long as the $\mathbf{\Lambda}$ and $\mathbf{\Sigma}$ (all diagonal) matrices in Eq. (7) are all positive, which they are. Example $\mathbf{\Lambda}_{\mathrm{h}c}$ and $\mathbf{\Lambda}_{\mathrm{h}\rho}$ spectra are shown in Fig. 2b.

- Point 7: *Lines 169-172 Spectral methods and spherical harmonics don't work great for atmospheric chemistry problems because the variability of chemical species does not follow wave structures ...*

    – The spherical harmonics are just a linear basis to represent the fields, so can in principle represent any form that the grid can represent. It is not true therefore that wave structures are the only kinds of structures that can be represented. The concept that is exploited in the work is that the a-priori errors are uncorrelated between different wavenumbers. This naturally translates to homogeneous correlation functions in real space (and isotropic if the error variances are assumed a function of total wavenumber only as is done here).

- Point 8: *Line 218 'weak sink over Antarctica'. Where does this sink come from?*

    – This appears to be an artifact of the plotting code; in reality there are no a-priori methane fluxes over Antarctica. Apologies for this. We have found the problem and will produce new figures in a revision. The fix is unlikely to affect other plots, but we will check this.

- Point 9: *Line 226 errors in atmospheric chemistry problems generally do not show 'homogeneous and isotropic correlations'.*

    – This is the case in many geophysical systems, where data assimilation assumes homogeneous and isotropic correlation shapes (e.g. in many of the variational systems currently used for flux inversion, apart from at land-sea boundaries). It is an assumption that follows from the spectral method used in the paper, but as stated in first bullet point above, the land and sea points can be decoupled by using two spectral control variables and masks. It is also true to say, for instance, that atmospheric chemistry problems do not obey Gaussian statistics, but this does not make the Gaussian assumption inapplicable or useful (most atmospheric inversion systems assume Gaussian statistics). We believe the work in the paper is a reasonable first piece of work to test the spectral method for the problem of flux inversion.

- Point 10: *Fig. 2 The correlation length scale looks more like ten degrees, so ~1000 km. Seems long, particularly for methane fluxes which come from a diversity of uncorrelated source sectors. What is the rationale for a spatial error correlation in methane fluxes?*

    – A SOAR (Second Order Auto Regressive) function is used for the correlation functions in this work and are shown for length-scales of 400 km and 600 km in Fig. 2a. For a given length-scale parameter, $\ell$, the SOAR may be compared to exponential and Gaussian correlations functions – SOAR: $(1 + |x/\ell|) \exp(-|x/\ell|)$, exponential: $\exp(-|x/\ell|)$, Gaussian: $\exp(-(x/\ell)^2/2)$. These are plotted in Fig. 1 below. For a given $\ell$ the SOAR 'looks' longer scale than the other two forms. This explains why the curves in Fig. 2a appear longer than the length-scales specified.

    – The reasons for adopting non-zero length-scales are many. (i) Even if emissions are physically uncorrelated, the *errors* between neighbouring locations can still be correlated (the correlation used in data assimilation rightly does not need to be associated with causation). (ii) A non-zero regularisation length-scale can be advantageous to produce a sufficiently smooth analysis. (iii) In the tests done in this paper, the results were consistently better when correlations were imposed.

- Point 11: *Figure 3 weird to have the vertical error correlation plot emphasize the stratosphere in a demo for methane fluxes.*

[Figure]

Fig. 1: Comparison of SOAR (Second Order Auto Regressive), exponential, and Gaussian correlation functions, all with $\ell = 600$ km.

– The vertical correlations in Fig. 3 are not for methane fluxes, but for the initial conditions of methane, which includes a representation in the stratosphere. Hence we think it is fair to show the stratospheric part.

• Point 12: *Line 284* (we assume the reviewer means Line 384) *The small information content to be obtained from the 60 NOAA stations is incommensurate to the size of the state vector. That would explain why the results are disappointing.*

– We don't regard the results as disappointing. As stated above, although there are fewer observations than used in a study using large amounts of satellite data, the data are still useful and effective for our purposes.